# Distributed representations of prediction error signals across the cortical hierarchy are synergistic

Frank Gelens[1,2,11], Juho Äijälä[2,11], Louis Roberts[2,3], Misako Komatsu [4], Cem Uran[5,6], Michael A. Jensen [7], Kai J. Miller[7], Robin A. A. Ince [8], Max Garagnani [3,9], Martin Vinck[5,6] ✉ & Andres Canales-Johnson [2,10] ✉

A relevant question concerning inter-areal communication in the cortex is whether these interactions are synergistic. Synergy refers to the complementary effect of multiple brain signals conveying more information than the sum of each isolated signal. Redundancy, on the other hand, refers to the common information shared between brain signals. Here, we dissociated cortical interactions encoding complementary information (synergy) from those sharing common information (redundancy) during prediction error (PE) processing. We analyzed auditory and frontal electrocorticography (ECoG) signals in five common awake marmosets performing two distinct auditory oddball tasks and investigated to what extent event-related potentials (ERP) and broadband (BB) dynamics encoded synergistic and redundant information about PE processing. The information conveyed by ERPs and BB signals was synergistic even at lower stages of the hierarchy in the auditory cortex and between auditory and frontal regions. Using a brain-constrained neural network, we simulated the synergy and redundancy observed in the experimental results and demonstrated that the emergence of synergy between auditory and frontal regions requires the presence of strong, long-distance, feedback, and feedforward connections. These results indicate that distributed representations of PE signals across the cortical hierarchy can be highly synergistic.

The traditional modular view of brain function is increasingly challenged by the finding that information about external stimuli and internal variables is distributed across brain areas[1-5]. When information in a complex system is carried by multiple nodes, this could imply that there is a large degree of redundancy in the information carried by the different nodes. That is, the whole is actually less than the sum of the parts. An alternative possibility, however, is that information is carried in a synergistic manner, i.e. the different nodes might carry extra information about task variables when they are combined. This can occur when the relationship between the nodes encodes the stimulus

[1]Department of Psychology, University of Amsterdam, Nieuwe Achtergracht 129-B, 1018 WT Amsterdam, The Netherlands. [2]Department of Psychology, University of Cambridge, CB2 3EB Cambridge, UK. [3]Department of Computing, Goldsmiths, University of London, SE14 6NW London, UK. [4]Laboratory for Haptic Perception and Cognitive Physiology, RIKEN Brain Science Institute, Saitama 351-0198, Japan. [5]Ernst Strüngmann Institute (ESI) for Neuroscience in Cooperation with Max Planck Society, 60528 Frankfurt am Main, Germany. [6]Donders Centre for Neuroscience, Department of Neuroinformatics, Radboud University Nijmegen, 6525 Nijmegen, The Netherlands. [7]Department of Neurosurgery, Mayo Clinic, Rochester, MN 55905, USA. [8]School of Psychology and Neuroscience, University of Glasgow, Glasgow G12 8QB, Scotland, UK. [9]Brain Language Lab, Freie Universität Berlin, 14195 Berlin, Germany. [10]Neuropsychology and Cognitive Neurosciences Research Center, Faculty of Health Sciences, Universidad Católica del Maule, 3460000 Talca, Chile. [11]These authors contributed equally: Frank Gelens, Juho Äijälä. ✉e-mail: martin.vinck@esi-frankfurt.de; afc37@cam.ac.uk

in a way that is not apparent when observing each node's activity alone —in other words, the whole is greater than the sum of the parts[6].

Both recent large-scale spiking and electrocorticographic (ECoG) recordings support the notion that information about task variables is widely distributed rather than highly localized[2,7–10]. For example, in the visual domain, widespread neuronal patterns across nearly every brain region are non-selectively activated before movement onset during a visual choice task[7]. Similarly, distributed and reciprocally interconnected areas of the cortex maintain high-dimensional representations of working memory[10]. In the case of multisensory integration, sound-evoked activity and its associated motor correlate can be dissociated from spiking activity in the primary visual cortex (V1)[11,12]. A last example, and the one used in the current study, is the case of communication of prediction error (PE) signals. Hierarchical predictive coding theory has been proposed as a general mechanism of processing in the brain[13]. The communication of prediction error (PE) signals using spikes and local field potentials (LFPs) recorded from subcortical and cortical regions reveal a large-scale hierarchy of PE potentials[8].

A major question is whether such distributed signals exhibit a high degree of redundancy (i.e. shared information) or a high degree of synergy (i.e. extra information) about their corresponding task variables. Electrophysiological studies have shown that synergy and redundancy have functional relevance[6,14–19]. For instance, laminar recordings in V1 suggest that synergistic interactions can efficiently decode visual stimuli better than redundant interactions, even in the presence of noise and overlapping receptive fields[14]. In contrast, the information processing of olfactory stimuli exhibits higher levels of redundant information across olfactory regions[20]. Here we investigate this question by using co-Information (co-I), an information theoretical metric capable of decomposing neural signals into what is informationally redundant and what is informationally synergistic about a stimuli variable[15,21]. Redundant information quantifies the shared information between signals, suggesting a common processing of the stimuli. Synergistic information quantifies something different: whether there is extra information only available when signals are combined, indicating that the information about the variable is in the actual relationship between the signals. Using ECoG recordings, we investigated synergistic and redundant interactions in five common marmosets performing two types of auditory tasks. This allowed us to determine the processing of communication of prediction error information across the brain during a range of auditory deviancy effects. Finally, to investigate how structural connectivity might facilitate the emergence of synergy, we applied the same oddball stimulation task used in the experiments to a brain-constrained neurocomputational model of the relevant cortical areas[22,23]. We computed synergy and redundancy in the simulated responses while manipulating the network's connectivity structure to unravel a potential/candidate mechanism responsible for generating the synergistic interactions observed in vivo (Fig. 1).

## Results
### Mutual information reveals prediction error effects within cortical areas
To characterize the distribution of PE across multiple cortical areas, we quantified PE in each electrode of the five marmosets by contrasting deviant and standard tones (Fig. 2). For each electrode, we computed Mutual Information (MI) to quantify the relationship between tone category (standard vs deviant) with their corresponding ECoG signal across trials. Within the framework of information theory, MI is a statistical quantity that measures the strength of the dependence (linear or non-linear) between two random variables. It can be also seen as the effect size, quantified in bits, for a statistical test of independence[15]. Thus, for each electrode and time point, we considered ECoG signals corresponding to standard and deviant trials and utilized MI to quantify the effect size of their difference.

We have recently proposed that a suitable candidate for broadcasting unpredicted information across the cortex is the transient, aperiodic activity reflected at the level of the event-related potentials (ERP) and broadband power (BB)[24]. A well-studied ERP marker of auditory PE is the mismatch negativity (MMN), an ERP that peaks around 150–250 ms after the onset of an infrequent acoustic stimulus[8,25–27]. A second neural marker of auditory PE is the BB response, an increase in spectral power spanning a wide range of frequencies usually above 100 Hz[27,28]. Whereas ERPs reflect a mixture of local potentials and volume conducted potentials from distant sites, BB is an electrophysiological marker of underlying averaged spiking activity generated by the thousands of neurons that are in the immediate vicinity of the recording electrodes[29,30]. MI was computed separately for the two neural markers of prediction error (i.e. ERP and BB signals). Electrodes showing significant differences in MI over time (see Methods) are depicted in Fig. 2. In the Roving Oddball Task, ERP signals showed PE effects across multiple cortical regions not necessarily restricted to canonical auditory areas (Fig. 2b). In the case of the BB signal, MI analyses revealed PE effects located predominantly in the auditory cortex of the three marmosets, as well as in a few electrodes located in the frontal cortex of marmoset Kr and Go (Fig. 2a). These results agree with previous studies in different sensory modalities[29] showing that broadband responses are spatially localized. In the case of the Local/Global Task, although the dataset for marmoset Nr and Ji contained ECoG recording only from the temporal and frontal cortices, the overall PE effects in the ERP signals were observed in a higher number of electrodes than in the BB signal (Fig. 2 and Fig. S9).

### Co-Information reveals redundant and synergistic cortical interactions
To investigate how auditory PE signals are integrated within and between the cortical hierarchy, we quantified redundant and synergistic cortical interactions using an information-theoretic measure known as co-Information (co-I)[15]. Co-I quantifies the type of information that interacting signals encode about a stimuli variable: positive co-I indicates redundant interactions between signals; and negative co-I accounts for synergistic interactions (Fig. 1d). Redundancy implies that the signals convey the same information about PE, indicating a shared encoding of PE information across time or space from trial to trial. On the other hand, synergy implies that signals from different time points or areas convey extra information about PE only when considered together, indicating that the relationship itself contains information about PE that is not available from either of the signals alone (Fig. 1d).

To quantify the dynamics of redundancy and synergy temporally and spatially, we computed the co-I within and between cortical areas (see Methods). We analyzed ERP and BB markers of PE separately, focusing our contrasts on the electrodes that showed significant MI effects in the analyses described in Fig. 2c.

### Experiment 1: Roving oddball task
**Temporal synergy and redundancy.** The finding that multiple recording sites encode information about PE raises the question of whether these signals convey the same or complementary PE information over time within a cortical region. Thus, we first characterized synergistic and redundant temporal interactions within ERP and BB signals. In the Roving Oddball Task, co-I analyses revealed widespread temporal clusters of synergistic information (in blue) and redundant information (in red) across the three monkeys in the auditory cortex (Fig. 3a, b), and frontal cortex (Fig. 3c, d). The ERP signal in the auditory (Fig. 3a) and frontal (Fig. 3c) electrodes showed characteristic off-diagonal synergistic patterns, resulting from the interaction between early and late time points within the same ERP signal (e.g. Fig. 3a, c; grey clusters

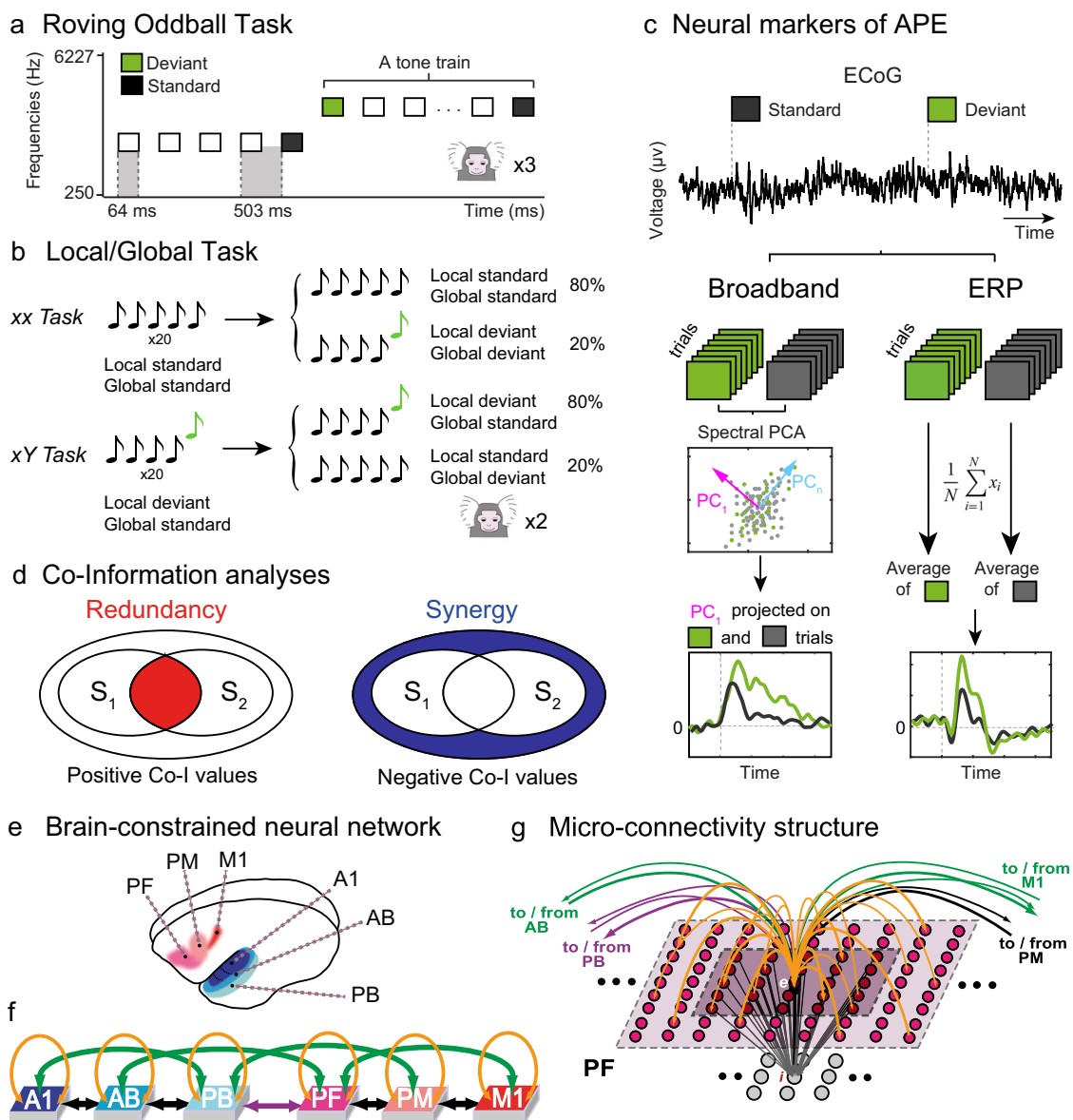

**Fig. 1 | Experimental design, information-theory analyses, and modelling.**
**a** Using a Roving oddball Task, 20 different single tones were presented in the trains of 3, 5, or 11 identical stimuli. Any two subsequent trains consisted of different tones. This way, while the adjacent standard (depicted in black) and deviant (depicted in green) tones deviated in frequency due to the transition between the trains, the two expectancy conditions were physically matched, as the first and the last tones of the same train were treated as deviant and standard tones in the analysis of the adjacent stimuli pairs. This task was performed by 3 marmosets (Fr, Kr, and Go). **b** Local/Global Task. On each trial, five tones of 50-ms-duration each were presented with a fixed stimulus onset asynchrony of 150 ms between sounds. The first 4 tones were identical, either low-pitched (tone A) or high-pitched (tone B), but the fifth tone could be either the same (AAAAA or BBBBB, jointly denoted by xx) or different (AAAAB or BBBBA, jointly denoted by xY). Each block started with 20 frequent series of sounds to establish global regularity before delivering the first infrequent global deviant stimulus. This task was performed by 2 different marmosets (Ji and Nr). **c** Neural markers of auditory prediction error. Deviant (green) and standard (black) epochs are used to compute the broadband and ERP responses. Broadband is computed by extracting by reconstructing the time series of standard and deviants with the first spectral principal component (SPCA) of the ECoG signal; ERPs are computed by averaging the raw voltage values for standard

and deviant trials (see "Methods"). **d** Schematic representation of redundancy and synergy analyses computed using co-Information. Each inner oval (A1 and A2) represents the mutual information between the corresponding ECoG signals and the stimuli category (standard or deviant). The overlap between A1 and A2 represents the redundant information about the stimuli (red; left panel). The outer circle around A1 and A2 represents the synergistic information about the stimuli (blue; right panel). **e** Brain areas modelled, network architecture, and its connectivity. Top left: Cortical areas modelled. Three cortices in the left temporal lobe (primary auditory: A1, auditory belt: AB, and parabelt: PB) are involved in auditory processing, and three in the frontal lobe (prefrontal: PF; premotor: PM; primary motor: M1) directly linked to them. **f** Network architecture. All the (sparse and random) connections are based on marmoset neuroanatomy (see Methods). **g** Schematic of links to/from a single excitatory cell '*e*'. Each model area consists of two layers of excitatory (upper) and inhibitory (lower) graded-response leaky integrator cells with neuronal fatigue. Dense links between these layers (grey arrows) implement mutual inhibition between (**e**) and its neighbors. Panels (**a** and **b**) are adapted from ref. 26, under a CC-BY license: https://creativecommons.org/licenses/by/4.0/.
Panels (**f** and **g**) are adapted from ref. 22, Copyright Elsevier (2013) under a CC-BY 3.0 license: https://creativecommons.org/licenses/by/3.0/. Panel (**e**) is adapted from ref. 28, under a CC-BY license: https://creativecommons.org/licenses/by/4.0/.

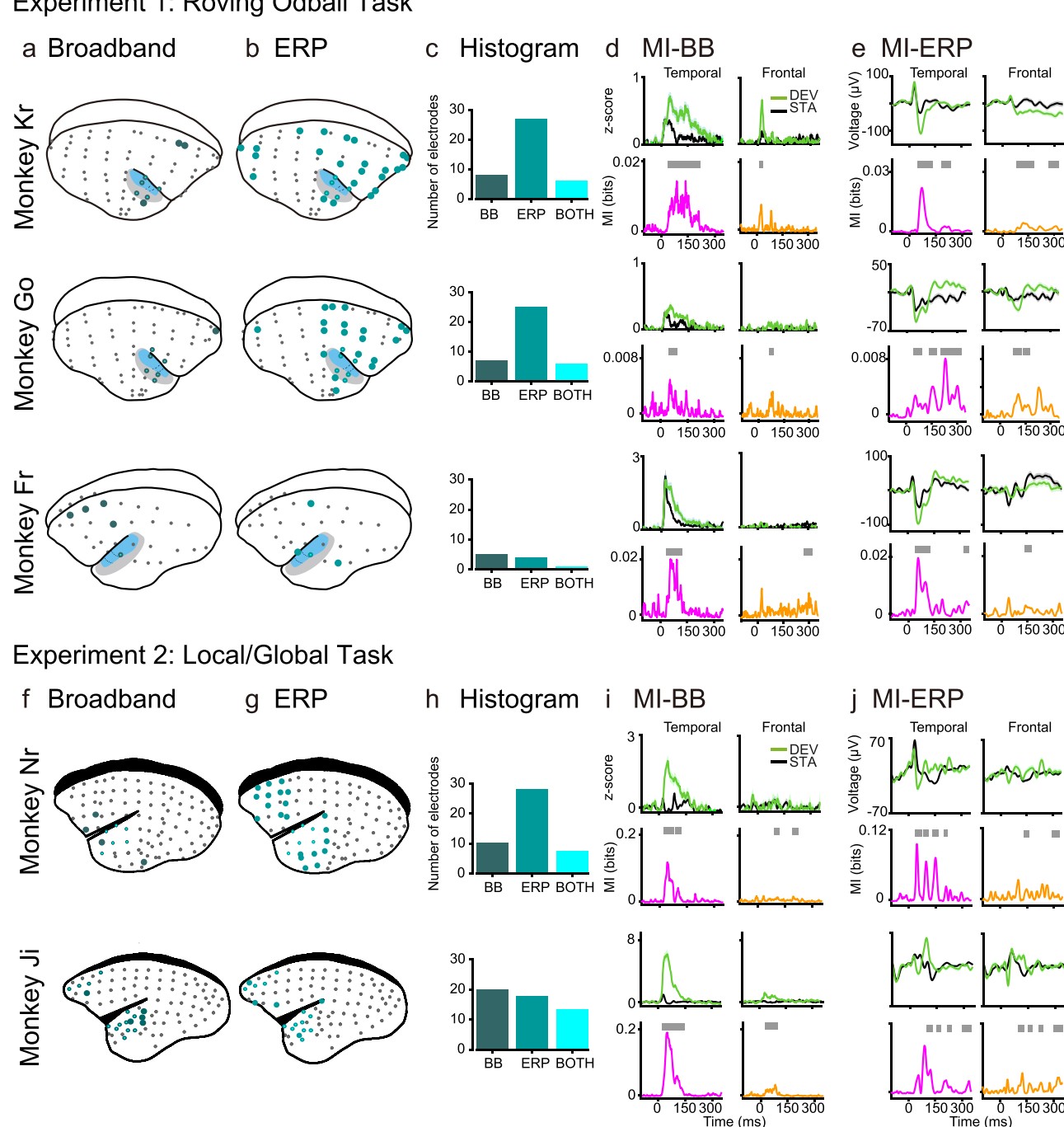

**Fig. 2 | Broadband and ERP markers of PE across the monkey brain.** Electrode locations for marmoset Kr (64 electrodes), Go (64 electrodes), and Fr (32 electrodes) in Experiment 1; and Nr (96 electrodes in ECoG-array, 39 used for analyses) and Ji (96 electrodes in ECoG-array, 27 used for analyses) in Experiment 2. Electrodes showing significant PE effect after computing MI between standard and deviant trials for the (**a**, **f**) Broadband (dark green circles) and (**b**, **g**) ERP (light green circles) markers of auditory prediction error. Electrodes showing significant MI for both markers are depicted in cyan. **c**, **h** Histogram of electrodes showing significant MI between tones for BB (left), ERP (middle), and both markers (right) for each animal. **d**, **i** Electrodes with the highest MI in the temporal and frontal cortex showing the BB signal for deviant and standard tones. Deviant tone (green) and

standard tone (black), and the corresponding MI values in bits (effect size of the difference) for the temporal (pink trace) and frontal (orange trace) electrodes. Significant time points after a permutation test are shown as grey bars over the MI plots. **e**, **j** Electrodes with the highest MI in the temporal and frontal cortex showing the ERP signal for deviant and standard tones. Error bars indicate standard error of the mean (SEM) across trials. For MI curves, we applied one-sided non-parametric permutation tests, correcting for multiple comparisons with the method of maximum statistics (see Methods). Grey bars show significant time windows with FWER $p < 0.05$. Panels (**a** and **b**) are adapted from ref. 26, under a CC-BY license: https://creativecommons.org/licenses/by/4.0/. Panels (**f** and **g**) are adapted from ref. 28, under a CC-BY license: https://creativecommons.org/licenses/by/4.0/.

between -140–300 ms after tone presentation), and revealed by the single electrode contrast depicted in Fig. S1. We observed significant temporal redundancy in the auditory (Fig. 3b) and frontal (Fig. 3d) BB signals. For auditory BB signals, the dynamics of the

redundant patterns were observed along the diagonal of the co-I chart, they were sustained over time and observed between time points around the early MI peaks (i.e., during the transient period when the effect sizes are larger between tones) (Fig. 3b; grey

# Roving Task

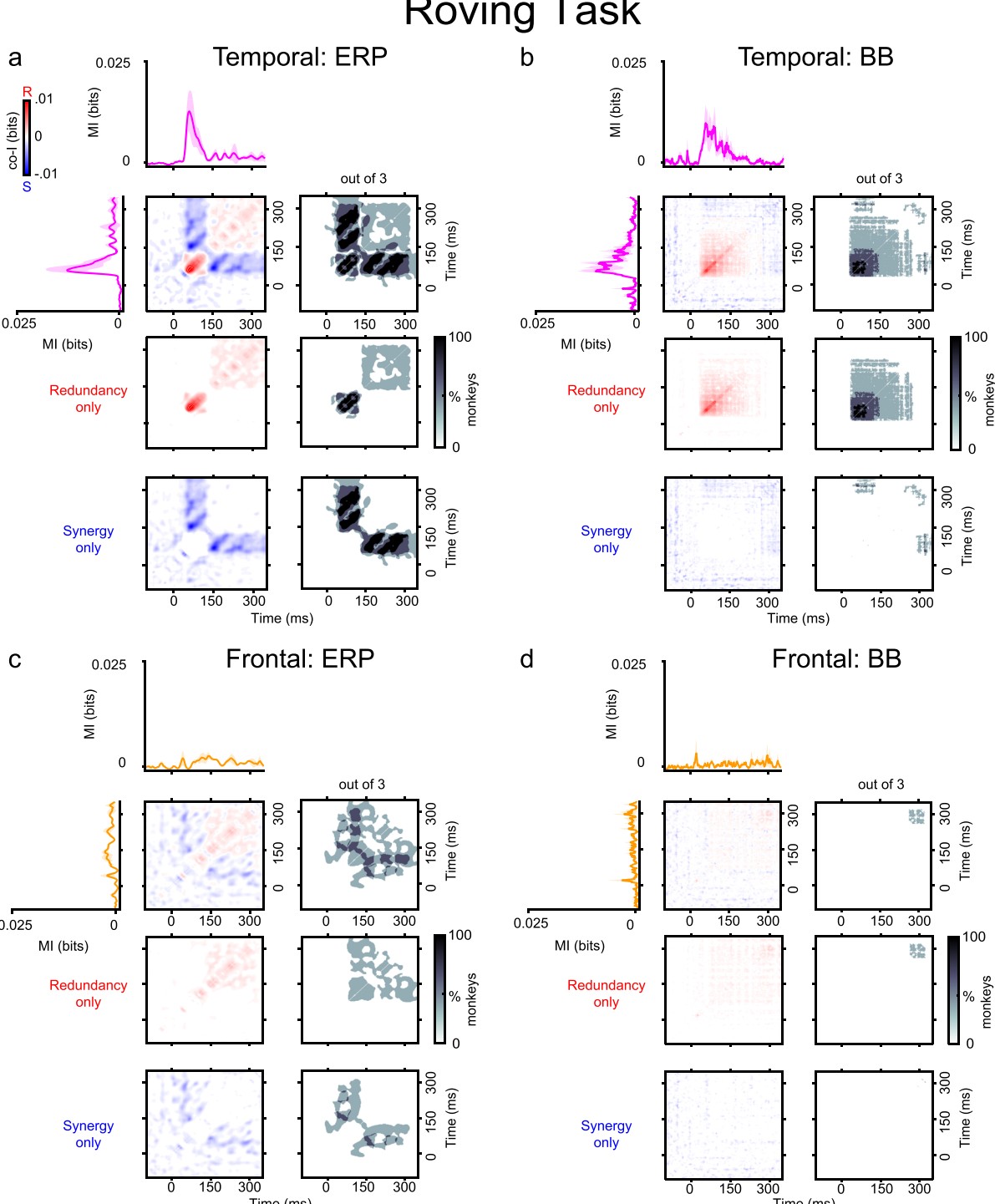

**Fig. 3 | Temporal synergy and redundancy within ERP and BB signals in the auditory and frontal electrodes with the highest MI for the Roving Oddball Task (experiment 1).** Co-information revealed synergistic and redundant temporal patterns within ERP (Panel **a**) and BB (Panel **b**) signals in the auditory cortex, and within ERP (**c**) and BB (**d**) signals in the frontal cortex. MI (solid traces) between standard and deviant trials for auditory (pink color) and frontal (orange color) electrodes averaged across the three monkeys. Error bars indicate SEM across electrodes. Temporal co-I was computed within the corresponding signal (ERP, BB) across time points between −100 and 350 ms after tone presentation. The average of the corresponding electrodes across monkeys is shown for the complete co-I chart (red and blue plots); for positive co-I values (redundancy only; red panel); and negative co-I values (synergy only; blue plot). The grey-scale plots show the proportion of monkeys showing significant co-I differences in the single electrodes analysis depicted in Fig. S1. Source data are provided as a Source Data file.

clusters -120–280 ms after tone presentation). In the frontal electrodes, we observed significant clusters of sustained redundant interactions around later time points (Fig. 3d; grey cluster around 300 ms after tone presentation).

**Spatio-temporal synergy and redundancy.** The finding that multiple recording sites encode information about PE raises the question of whether these regions are dynamically interacting and whether these inter-areal interactions are redundant or synergistic. To test this

# Roving Task

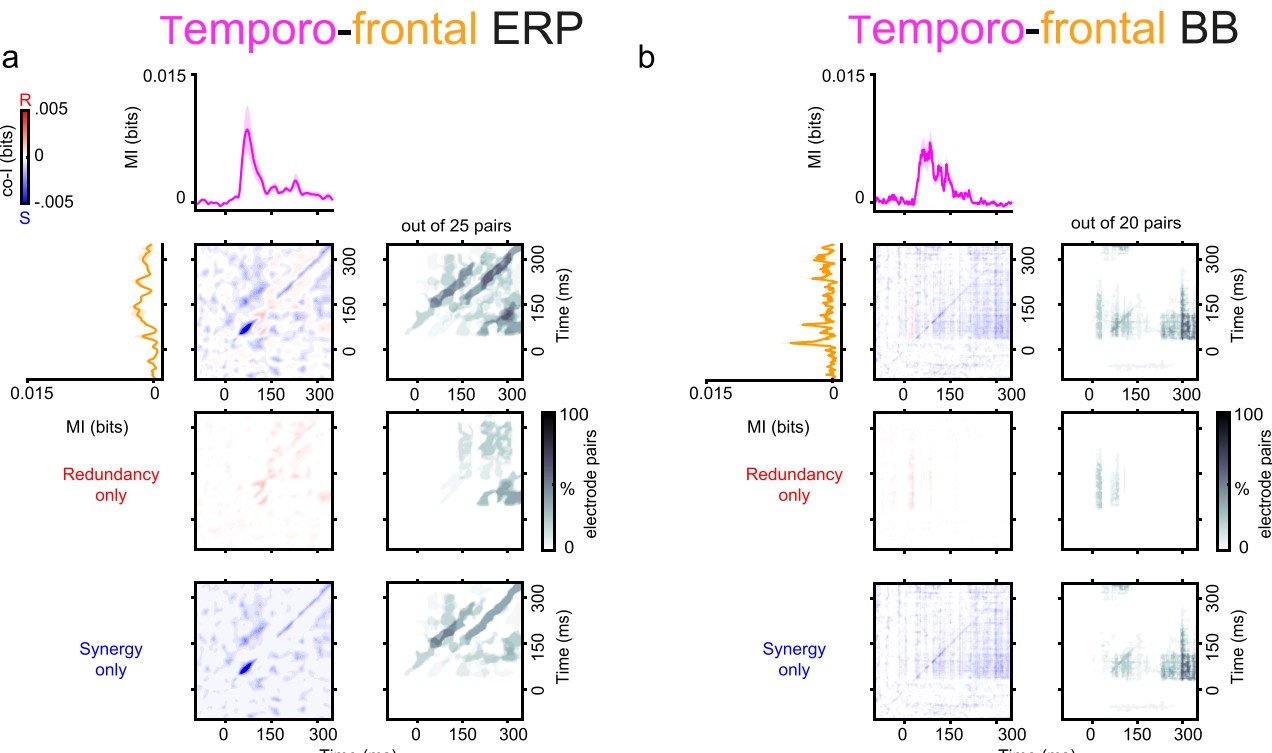

**Fig. 4 | Spatio-temporal synergy and redundancy between auditory and frontal electrodes in the Roving Oddball Task (experiment 1).** Co-information revealed synergistic and redundant spatio-temporal patterns between auditory and frontal electrodes in the ERP (Panel **a**) and BB (Panel **b**) signals for the Roving Oddball Task. MI (solid traces) between standard and deviant trials for temporal (pink color) and frontal (orange color) electrodes. Error bars indicate SEM across electrodes. Co-I was computed between each pair of electrodes and across time points between −100 and 350 ms after tone presentation. The average of the temporo-frontal pairs across the three monkeys is shown for the complete co-I chart (red and blue plots); for the positive co-I values (redundancy only; red plot); and the negative co-I values (synergy only; blue plot). The proportion of electrode pairs showing significant co-I differences is shown in the corresponding grey-scale plots. The average co-I charts for the individual monkeys are shown in Fig. S3 for the ERP signals and in Fig. S6 for the BB signals. Source data are provided as a Source Data file.

possibility, we characterized the redundancy and synergy between auditory and frontal electrodes. Spatio-temporal co-I was computed between the auditory and frontal electrodes over time (Fig. 4) and averaged across monkeys separately in the Roving Oddball Task (i.e. ERP and BB signals). The dynamics of spatio-temporal synergy in the ERP and BB signals showed complex and heterogeneous patterns between early time points of the auditory electrodes and later time points in the frontal electrodes (Fig. 4). For example, while the ERP signals encoded both diagonal (Fig. 4a; grey clusters -100–350 ms after tone presentation) and off-diagonal synergistic patterns (Fig. 4a; grey clusters -150–350 ms after tone presentation), the BB signals mainly showed off-diagonal synergy between temporal and frontal electrodes (Fig. 4b; grey clusters -220–350 ms after tone presentation). In Fig. 4a, the diagonal stripes suggest the possibility of oscillatory dynamics, where the representation in frontal regions between 50 and 300 ms is enhanced by knowledge of the activity of temporal regions -50 ms earlier (the upper diagonal line). Note that 50 ms peak-to-peak timescale corresponds to a frequency of -10 Hz, i.e. the alpha range. In Fig. 4b the off-diagonal block suggests that the frontal representation of the stimulus between 20–120 ms initiates a state change: later temporal activity (200 ms+) enhances the readout of the stimulus class, even though there is no representation of PE in the BB signal of the temporal area at that time.

## Experiment 2: Local/global task

**Temporal synergy and redundancy.** Although PE processing has been widely studied using the Roving Oddball Task[27], the contribution of stimulus-specific adaptation (SSA) to the amplitude of the ERP response is usually considered a confounding factor in the isolation of PE[8]. For this reason, we also investigated synergy and redundancy in a separate task capable of attenuating the effects of SSA (i.e. the Local/Global Task). In the Local contrast, although we observed temporal synergy in both ERP and BB signals, the off-diagonal synergy was primarily observed between early and late time points of the BB signals in the temporal cortex (Fig. 5; grey clusters -150–350 ms after tone presentation). The ERP signals, on the other hand, showed diagonal synergy in both the temporal (Fig. 5; grey clusters -40–150 ms after tone presentation) and frontal cortex (Fig. 5; grey clusters -150–350 ms after tone presentation).

Another advantage of the Local/Global Task is the possibility of exploring a higher-order PE observed as a violation of the overall sequence of tones (Global contrast; see Fig. 1b, and Methods). This context-dependent deviancy effect has been shown to elicit neural activation in frontal regions[28,31]. In the case of the Global contrast, we observed temporal synergy across early and late time points but mostly in the BB signals both within the auditory (Fig. 5f; grey clusters -0–350 ms after tone presentation) and frontal electrodes (Fig. 5H; grey clusters -230–330 ms after tone presentation). Taken together, these results suggest that the Local/Global Task elicits distributed patterns of PE information across time that are primarily encoded by firing rates.

**Spatio-temporal synergy and redundancy.** We investigated whether local and higher-order PE are encoded by synergistic information

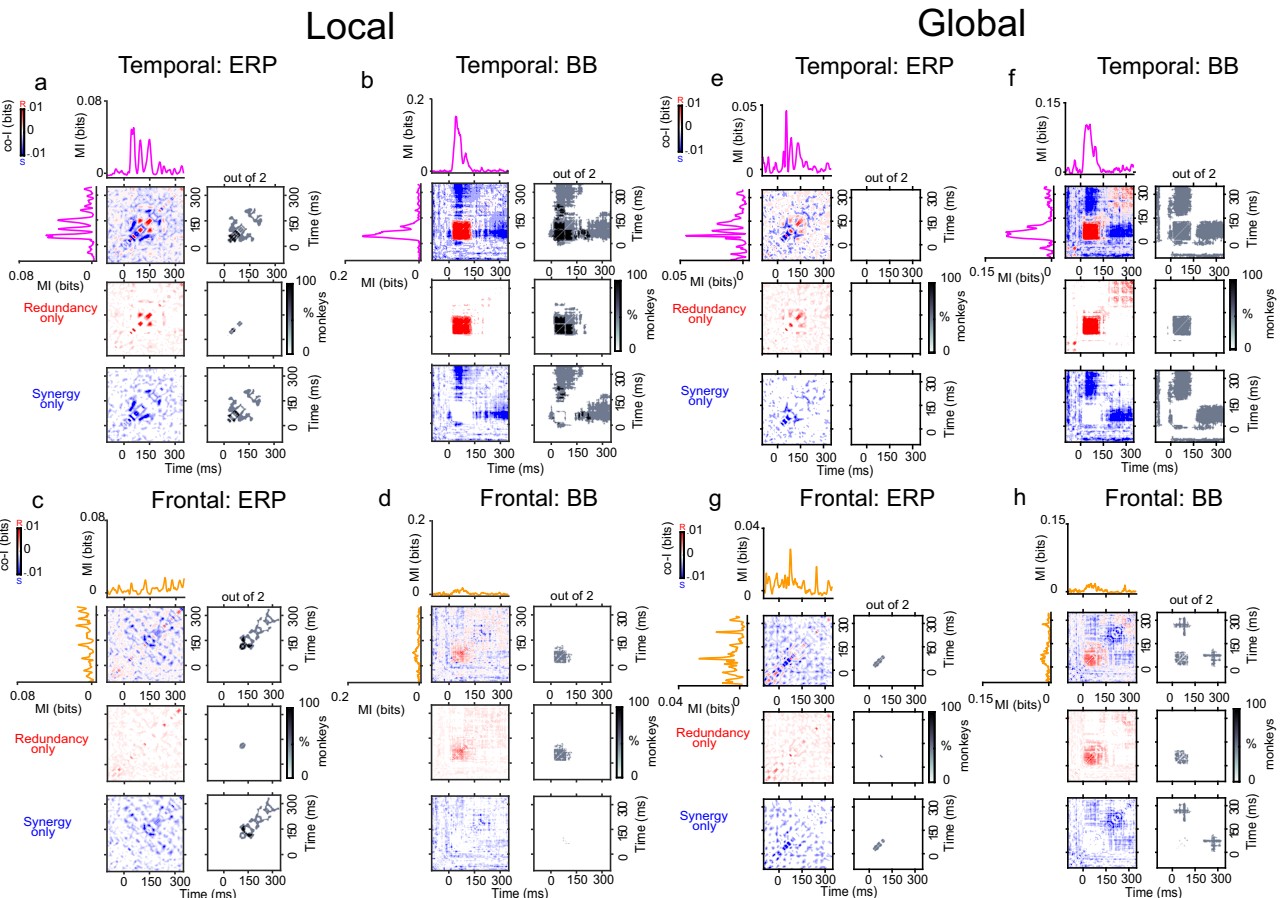

**Fig. 5 | Temporal synergy and redundancy within ERP and BB signals in the auditory and frontal electrodes with the highest MI for the Local/Global Task (experiment 2).** In the Local and Global contrasts, co-information revealed synergistic and redundant temporal patterns within ERP (Panels **a**, **e**) and BB (Panels **b**, **f**) signals in the auditory cortex, and within ERP (Panels **c**, **g**) and BB (Panels **d**, **h**) signals in the frontal cortex. MI (solid traces) between standard and deviant trials for auditory (pink color) and frontal (orange color) electrodes averaged across the three monkeys. Error bars indicate SEM across electrodes.

Temporal co-I was computed within the corresponding signal (ERP, BB) across time points between −100 and 350 ms after tone presentation. The average of the corresponding electrodes across monkeys is shown for the complete co-I chart (red and blue plots); for positive co-I values (redundancy only; red panel); and negative co-I values (synergy only; blue plot). The grey-scale plots show the proportion of monkeys showing significant co-I differences in the single electrodes analysis depicted in Fig. S2. Source data are provided as a Source Data file.

between cortical regions. Thus, we characterized the synergy (and redundancy) between auditory and frontal electrodes for the local and global contrast (Fig. 6). Spatio-temporal co-I was computed between the auditory and frontal electrodes over time and averaged across two monkeys separately for each signal (i.e. ERP and BB signals). Consistent with the effects observed in the Roving Oddball Task, we observed multiple patterns of synergistic and redundant information between temporal and frontal regions. We also noticed an interesting difference between the two tasks. While the Roving Oddball Task elicited most of the synergistic interactions between ERP signals (Fig. 4a), the Local/Global Task elicited most of the synergy between BB signals (Fig. 6b, d).

**Multivariate co-information (MVCo-I): synergy and redundancy.** Although the per-electrode and electrode-pair analyses of synergy and redundancy exploit the optimal spatial resolution of the recording modality across temporal and frontal regions, they could also miss information encoded in the spatial pattern both within and between temporal and frontal areas. They could therefore potentially miss synergy or redundancy that is only apparent when considering multiple electrodes together, either due to a low signal-to-noise ratio within each channel or because of a genuinely distributed informative spatial pattern. This might be particularly

relevant for the ERP signals that showed extensive temporal and frontal PE effects (Fig. 1a, b). Thus, to account for potential informative spatial patterns of redundancy and synergy in ERP signals, and to reduce any concern about high-order interactions between channels within each region in the pairwise channel analysis, we complemented our analyses by computing co-I based on the response across multiple electrodes (MVCo-I: Multivariate Co-information) (Figs. S11, S12). In brief, we have applied a cross-validated multivariate analysis approach that uses machine learning to capture the best linear representation of the prediction error signal across a whole region, and we have repeated our co-information analyses within and between the two brain regions of interest using the classifiers' outputs (frontal and temporal) (see Methods).

The MVCo-I analyses within-region (Figs. S13, S15) and between-regions (Figs. S14, S16) showed comparable co-I in terms of synergistic and redundant dynamics observed in the per-electrode (Figs. 3, 5) and in the between-electrodes (Figs. 4, 6) analyses, but with increased statistical power (i.e., increased MI).

To sum up, we observed widespread patterns of synergy within and between ERP and BB signals across the auditory cortical hierarchy. The distributed nature of the temporal and spatio-temporal synergistic interactions across the hierarchy raises the question of whether the

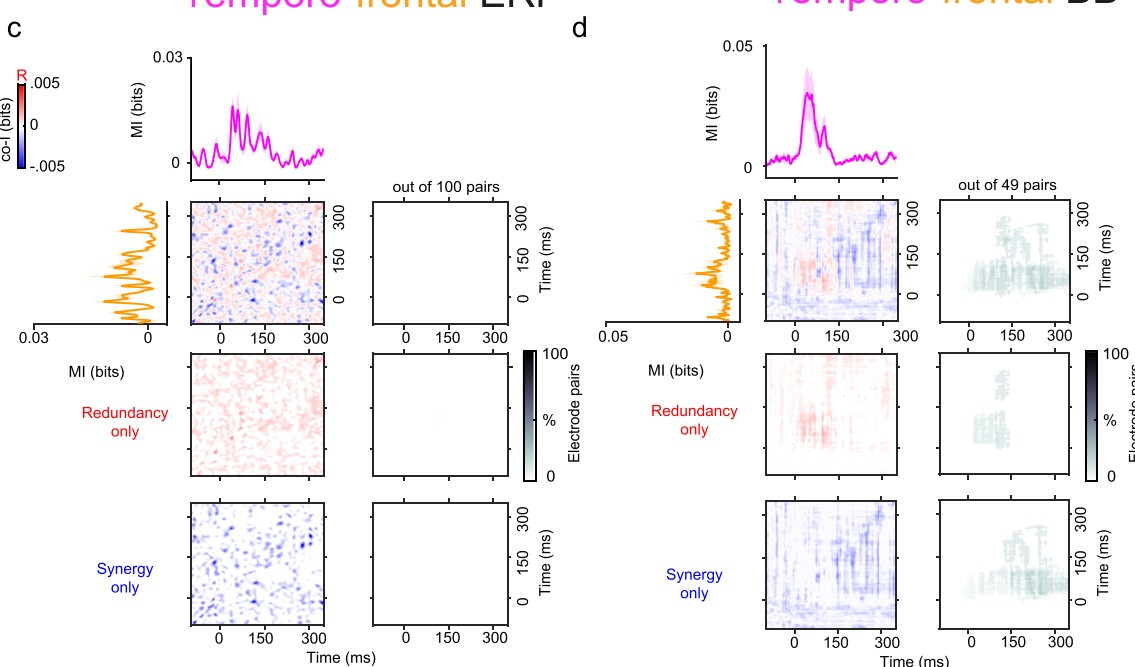

**Fig. 6 | Spatio-temporal synergy and redundancy between auditory and frontal electrodes in the Local/Global Task (experiment 2).** Co-information revealed synergistic and redundant spatio-temporal patterns between auditory and frontal electrodes in the ERP (Panels **a**, **c**) and BB (Panels **b**, **d**) signals. MI (solid traces) between standard and deviant trials for temporal (pink color) and frontal (orange color) electrodes. Error bars SEM across electrodes. Co-I was computed between each pair of electrodes and across time points between −100 and 350 ms after tone presentation. The average of the temporo-frontal pairs across the three monkeys is shown for the complete co-I chart (red and blue panel); for the positive co-I values (redundancy only; red panel); and the negative co-I values (synergy only; blue panel). The proportion of electrode pairs showing significant co-I differences is shown in the corresponding grey-scale panels. The average co-I charts for the individual monkeys are shown in Figs. S4 and S5 For ERP signal, and Figs. S7 and S8 for the BB. Source data are provided as a Source Data file.

emergence of synergy is a consequence of the recurrent and feedback connections in the auditory network.

**Explaining the presence of synergy using a brain-constrained neurocomputational model.** To test the validity of our working hypothesis that synergistic information may be driven mainly by recurrent and feedback connections, we applied an existing neural network model[22,23] closely reproducing structural and functional properties of relevant areas in the superior-temporal and inferior-frontal lobes of the primate brain (Fig. 7a) to simulate auditory PE processing (see Methods). Our approach was to employ a fully brain-constrained neurocomputational model that accurately replicates critical neurobiological and neuroanatomical features of the mammalian cortex and to stimulate this model using the same Roving Oddball Task adopted in Experiment 1. The network simulated neuronal firing rates, which are the main contributors of the BB signals; we, therefore, honed our modelling efforts on reproducing the BB signals observed in Experiment 1 (Figs. 3b, d and 4b). The model we used (see Fig. 1f) has been previously applied to successfully simulate and explain automatic auditory change detection and the MMN response to familiar and unfamiliar sounds in the human brain[32] and several other phenomena in the domains of language acquisition and processing, attention, memory, and decision making[22,23,33–36]—see[37] for a recent review. Here, we recorded the network's responses (measured in each area as the sum of all cells' firing rates) to predefined random patterns simulating standard and deviant tones; following the same procedure used to process the experimental data (see Methods), we then analysed the resulting PE signals. We observed that, before any adjustment of its parameter values, the network already encoded both redundant and synergistic information, specifically, in the signal from its superior-temporal region (including areas A1, AB, PB). We then further constrained the model's dynamics by fine-tuning three of its parameters (i.e., the strength of the neuronal adaptation, the local inhibition, and the between-area links) so that the temporal and spatial features of synergistic information encoded in the simulated PE responses would closely resemble those we observed experimentally. This process of parameter tuning did not qualitatively change the network's responses but simply improved the fit of the responses with the observed data.

To quantify the similarity of the co-I values between the real and the simulated data, we computed the Structural Similarity Index (SSIM). The SSIM assesses the structural similarity between two images, with values ranging from 0 (dissimilar) to 1 (highly similar). Hence, we converted the co-I plots of the real and simulated data to images and computed SSIM between them. While the SSIM between simulated and experimental co-I was 0.74 (Fig. 7b versus Fig. 3b), the frontal cortex comparison showed an SSIM of 0.83 between simulated and experimental co-I (Fig. 7c versus Fig. 3d). Both values were statistically significant above the chance level ($p < 0.05$) after comparing them to a distribution of surrogate SSIM values. The surrogate distribution was obtained by computing the SSIM between the experimental co-I image and a shuffled version of the simulated co-I image and repeating this procedure 1000 times.

Having attained a good match between experimental and simulated data (e.g., compare Fig. 7 panels b and c, with Fig. 3 panels b and d, respectively), we then moved on to address the main question, i.e., whether the presence of feedback and recurrent links in the underlying neural network has a direct impact on the emergence of synergy. To investigate this, we directly manipulated the model's structural connectivity and analysed the effects of such manipulation on its responses to the same stimuli. Specifically, we used two types of architectures: first, a fully-connected model (FC), having connectivity as shown in Fig. 7a, i.e., including both feedforward and feedback between-area and recurrent within-area, connections. It is important to stress that such connectivity reflects the neuroanatomical links known

to exist between (and within) corresponding superior-temporal and inferior-frontal cortices in the macaque brain, as well as between the human's homologue cortical areas (see Methods for details). Second, we ran a set of feed-forward-only (FF) model simulations, in which we artificially cut all the feedback and recurrent links of the FC architecture, while maintaining all the feedforward ones intact. By *feedback links* here we refer to all the links in the model going from right to left in Fig. 7a (i.e., from area PB to A1, from area PF to AB, from PM to PB, from M1 to PF, and from each area to its left-hand side next-neighbour). For each of these two model types, we ran three distinct simulations. In each simulation, the projections linking any two areas (and any area to itself) were *sparse* and established at random, with the probability of any two cells being connected by a synapse decreasing with their (modelled) cortical distance. The weights of all the synapses were also set to a small random value comprised between 0 and 0.1 (see Methods for details).

Given this, we treated each simulation run as the model correlate of a single marmoset in Experiment 1 (Fr, Kr, and Go), as it was produced using a slightly different (random) variation of the same prototype network architecture (FC or FF). During each simulation, we generated and recorded 100 trials (50 deviants and 50 standards) and analysed the co-I within and between the resulting network responses in exactly the same way as in the experimental data. The results showed that the FC model showed highly synergistic interactions between temporal and frontal regions (Fig. 8a, b). Crucially for our hypothesis, we observed that the removal of all the between-area feedback projections and recurrent within-area links of the network entirely prevented the emergence of synergistic interactions between frontal and temporal model regions (see Fig. 8d). Additional simulations obtained with a version of the architecture containing just nearest-neighbour between-area feedback (and feedforward) links, along with recurrent ones (see Fig. S10) again failed to reproduce such synergistic interactions, indicating that it is not simply the presence of feedback projections, but specifically of *higher-order*, or so-called "jumping"[35], cortico-cortical links connecting non-adjacent areas of the processing hierarchy that is needed for synergy to emerge in the model.

## Discussion

In this study, we focused on computing temporally-resolved metrics of redundancy and synergy, aiming at investigating the dynamics of the information interactions within and between cortical signals encoding PE. Due to the interplay between temporal and spatial neural dynamics, our approach revealed a rich repertoire of redundant and synergistic patterns, showing transient and sustained information dynamics distributed across the auditory hierarchy.

There are a wide variety of ways of using information-theoretic, and other measures to study representational interactions in neural coding[38]. Schneidman et al.[39] discuss three types of response independence in the context of spiking neuron population coding: activity independence, conditional independence, and information independence. Here we focus only on information independence, as we are interested in relating the information representation between areas. Deviations from information independence are best measured with co-information. To date, co-information has been less frequently applied to aggregate signals as we do here[15,17,40].

There has recently been increasing interest in using information-theoretic measures quantifying redundancy and synergy to investigate functional connectivity between brain regions in resting-state data[6]. We highlight that a major difference here is we consider redundancy and synergy *about* an experimentally controlled external stimulus manipulation. Our approach therefore admits a more direct, grounded, interpretation, as synergy and redundancy are properties of distributed neural representations of a specific aspect of the animal's perception of the world (here the PE). Recent work in human MEG

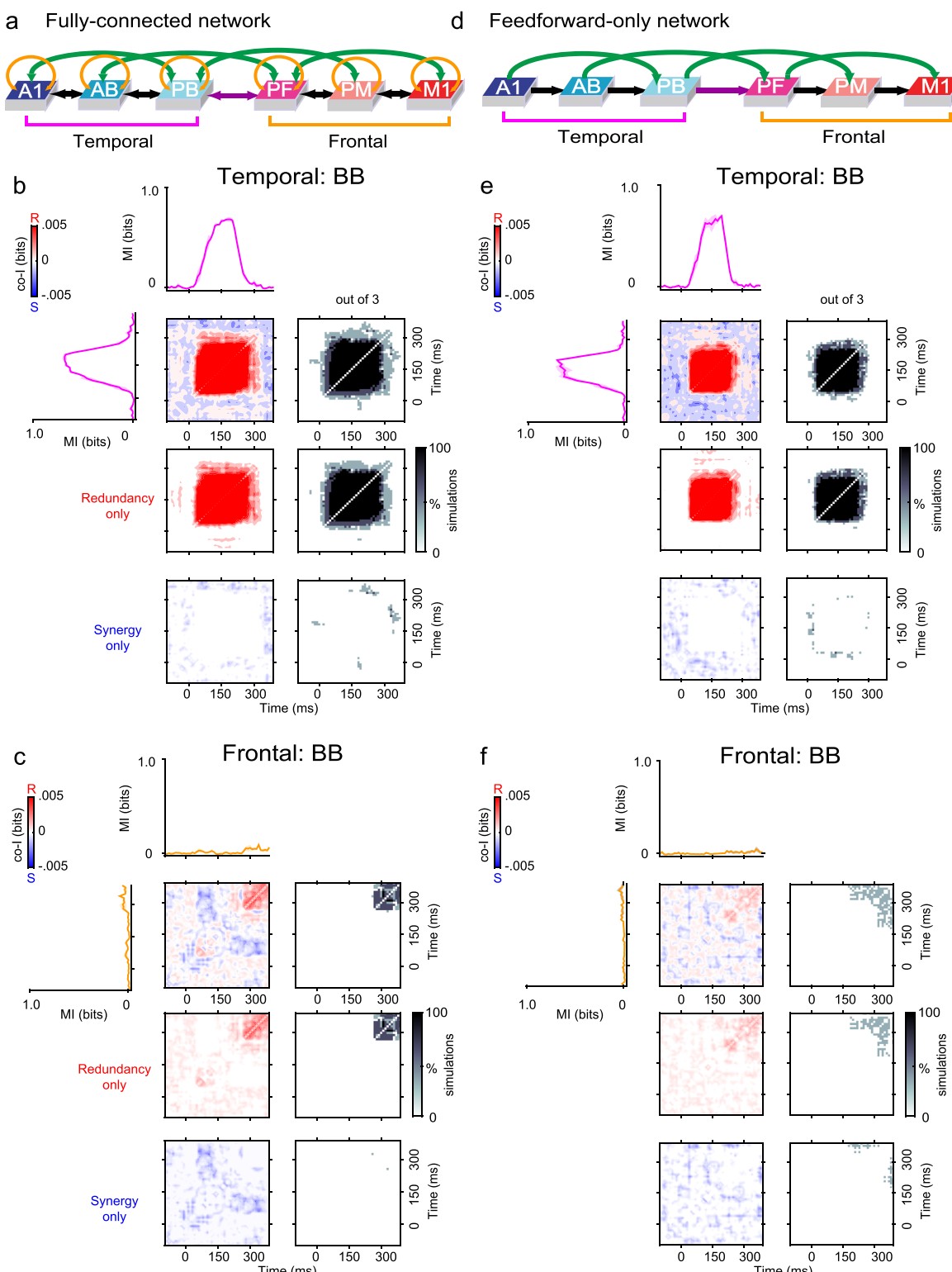

# Roving Task (Model)

**a** Fully-connected network

**d** Feedforward-only network

## Interpreting synergistic interactions

Synergistic information was observed mainly off-diagonally, i.e. between early and late times points after tone presentation for both within (Figs. 3–5) and between cortical areas (Fig. 6). This indicates that takes a similar approach and finds similar synergistic representations of PE[41].

late temporal responses carry information that, in combination with the early one, provides extra information about the identity of the tone (standard or deviant) than when considered in isolation. This raises the question about what is the functional relevance of synergistic information for representing prediction errors. The off-diagonal synergy between early and late time points could be a signature of a neural state shift. It is interesting to note that the synergy remains strong over

**Fig. 7 | Model architectures and simulation results.** A brain-constrained model of temporal and frontal areas of the marmoset brain (see Fig. 1f) was stimulated with simulated tones as in the Roving Oddball Task used in Experiment 1. **a, d** Different network architectures used for the simulations (see Methods). Feedforward and feedback between-area connections are depicted as black and green arrows; recurrent within-area links (panel a only) are shown in gold. Input stimuli were repeatedly presented to area A1 of the network (model correlate of primary auditory cortex) and firing rate responses of each excitatory cell within the six areas were recorded. **b, c** Results obtained with networks having a Fully Connected (FC) architecture (shown in panel **a**), which included both feedforward/feedback links and recurrent connections. **e, f** Results obtained using networks having a Feedforward-only (FF) architecture (panel **d**), in which the feedback and recurrent connections were absent. MI (solid traces) between standard and deviant trials averaged across three simulation runs (each run modelling a single monkey

dataset) are plotted for the three temporal (A1, AB, PB: pink curves) and three frontal (PF, PM, M1: orange curves) areas' simulated responses. Error bars represent SEM. Co-I analyses were performed on the model temporal and frontal areas' signals. Temporal co-I was computed within the simulated firing rates across time points between −100 and 350 ms after stimulus onset. The average of the corresponding electrodes across simulated monkey datasets is shown for the complete co-I chart (red and blue panel), for positive co-I values (redundancy only; red panel) and negative co-I values (synergy only; blue panel). The grey-scale panels show the proportion of simulated monkey datasets with the highest MI within the temporal (A1, AB, PB) and frontal (PF, PM, M1) regions. Note the similarity (in terms of temporal patterns of synergy and redundancy) between the results obtained from the FC model responses (panels **b** and **c**) and those from the corresponding experimental data (the BB signal shown in Fig. 3, panels **b** and **d**, respectively).

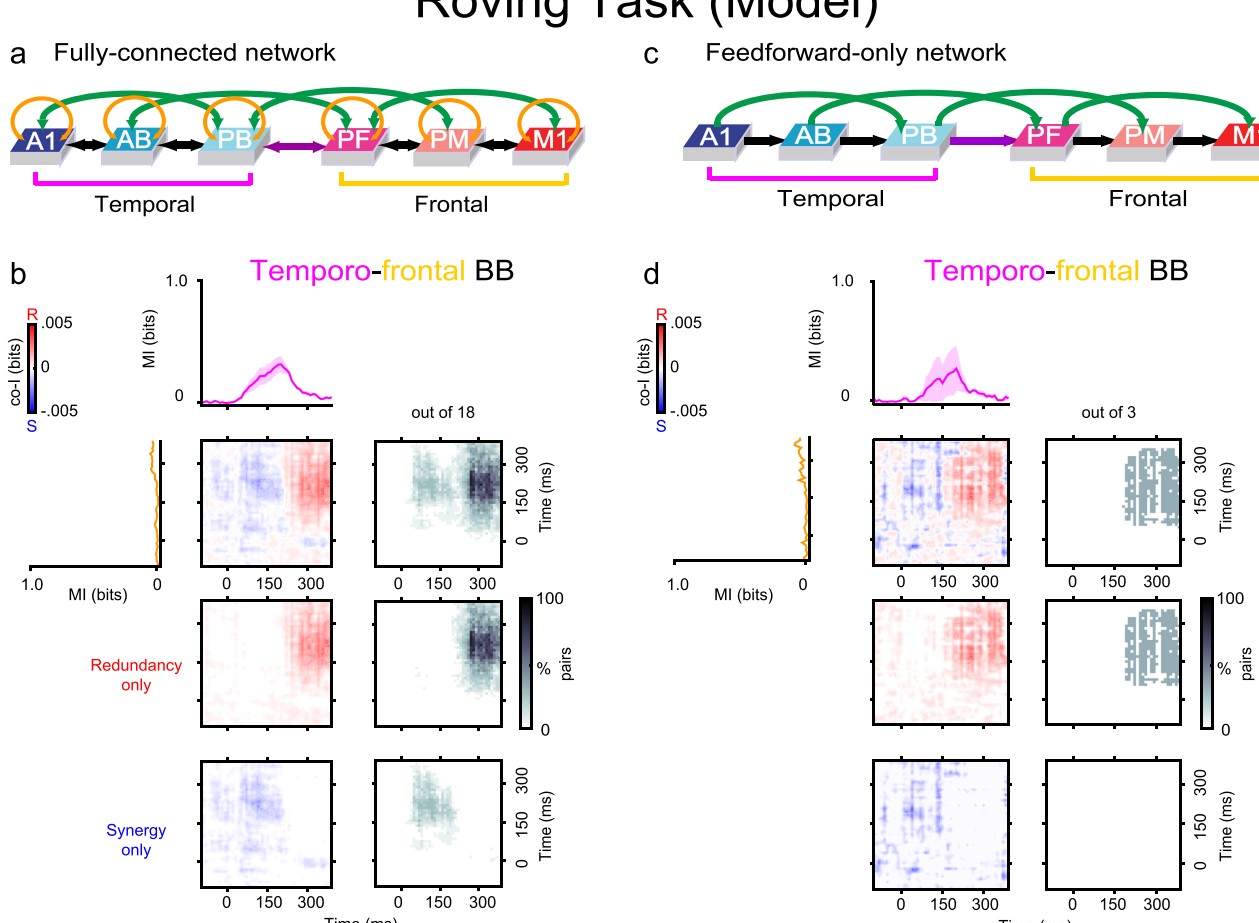

**Fig. 8 | Spatio-temporal synergy and redundancy of simulated signals.** The firing rate responses of the networks used to produce the results of Fig. 7 were subjected to co-I analyses between the simulated temporal and frontal areas' signals. **b** Results obtained using Fully Connected (FC) networks (panel **a**), which included both feedforward and feedback (black and green arrows) links and recurrent (golden arrows) connections. **d** Results obtained using Feedforward-only (FF) networks (panel **c**), in which the feedback and recurrent connections were absent (see Methods). MI (solid traces) between standard and deviant trials averaged across three simulation runs (each run modelling a single monkey dataset) are plotted for the three temporal (A1, AB, PB: pink curves) and three frontal (PF, PM, M1: orange curves) areas' simulated responses. Error bars indicate SEM. Co-I analyses were performed between the model temporal and frontal areas' signals. Temporal co-I was computed from the simulated firing rates across time points between −100 and 350 ms after stimulus onset. The average of the corresponding

electrodes across simulated monkey datasets is shown for the complete co-I chart (red and blue panel), for positive co-I values (redundancy only; red panel), and negative co-I values (synergy only; blue panel). The grey-scale panels show the proportion of significant co-I pairs between superior-temporal (A1, AB, PB) and frontal (PF, PM, M1) areas using areas that showed significant MI between standard and deviant trials. Note the similarity (in terms of spatio-temporal patterns of synergy and redundancy) between the results obtained from the model responses and those from the corresponding BB signals in the experimental data of Fig. 4: co-I measures of network responses show significant synergy between temporal and frontal regions (see panel **b**), as observed in real marmoset data (Fig. 4b). Also, note that such synergistic effects disappear after the removal of the network's feedback and recurrent links (compare the bottom-right plot of panel **b**, FC architecture, against that of panel **d**, FF architecture).

periods after the PE response is no longer represented (i.e. no MI at those time points). However, the initial representation of the PE may have shifted the local network dynamics into a different state. Then knowing this ongoing state improves the readout of the encoded information at the earlier time point. Thus, the off-diagonal synergy might be an echo of the initial PE representation that is not directly observable in later time points.

Synergy can also arise from a common source of neural noise that is non-stimulus specific. For example, the spatio-temporal synergy between regions could reflect a global change in attention or arousal. In this situation, the readout of one area provides information about the global neural state even when it does not convey information about the PE directly, and this can be used to improve the resolution with which the PE can be decoded from the other area. Although this might be a possibility, the tight timing of the synergy bands observed in both experiments (i.e., diagonal and off-diagonal synergistic patterns) speak more of a transient dynamics rather than global ongoing fluctuations underlying the spatio-temporal synergy.

How does synergistic information emerge from local and distributed neural dynamics? We reason that a plausible neurobiological mechanism for the generation of synergy is through recurrent and feedback neural interactions within and between areas, respectively. This hypothesis is corroborated by the novel neurocomputational results presented here. Specifically, we adapted a 6-layer-deep, brain-constrained neural network model reproducing the neuroanatomy and neurophysiology of language areas in the temporal and frontal cortex of the human brain[23,32,35] to simulate and explain the cortical mechanisms underlying the generation of the PE responses that we observed experimentally in the marmoset brain. In response to oddball stimulation with simulated auditory tones, the model was found to produce responses containing both synergistic and redundant information (here we looked at the network's per-area sum of all cells' firing rates; this measure can be related directly to the experimentally recorded BB signal). By tuning the model parameters, we were able to get the network's spatio-temporal patterns of synergy and redundancy encoding the PE response to closely replicate those found experimentally across the auditory cortical hierarchy; furthermore, manipulation of the model's connectivity revealed that synergistic interactions emerged in it only when strong, higher-order ("jumping") forward and backward links (green arrows in Fig. 1f) connecting frontal and temporal regions were present.

Based on this computational result, we conjecture that the cortical homologues of such jumping links (known to exist between corresponding regions of the marmoset brain; see Methods) may play a similarly crucial role in the emergence of the temporo-frontal synergistic interactions observed in the ECoG data. This prediction awaits further validation through experimental testing.

## Interpreting redundant interactions

A different type of dynamics was observed in the case of the redundant information across the cortex. Redundant patterns of information were observed mainly at time points close to the diagonal of the co-I chart, both within signals (Figs. 3–5) and between signals (Fig. 6). The advantage of computing redundancy is that it reveals to which extent local and inter-areal signals represent the same information about the stimuli category on a trial-by-trial basis. Redundant interactions about tone category (i.e., deviant or standard) were observed in the ERP and BB signals and represented the outcome of the shared information across time points (temporal redundancy) and between areas (spatio-temporal redundancy). These observed redundancy patterns raise the question of what is the functional relevance of redundant information for processing PE across the cortex.

A neurobiological interpretation of redundancy is that the neural populations encoding this type of information share a common

mechanism[15]. From the perspective of cortical dynamics, redundancy then could provide cortical interactions with robustness[6,20], as redundant interdependencies convey information that is not exclusive to any single cortical region. Robustness, understood as the ability to tolerate perturbations that might affect network functionality[6], is a desirable characteristic of cortical networks processing predictions to preserve stimuli separability in the presence of highly variable stimuli features, environmental noise, or endogenous sources of noise such as background neural activity. Thus, our results suggest that redundancy quantifies the robustness of the information processing in the cortex, enabling multiple areas to process common information about prediction errors. One way this could arise is from multiple regions receiving the same input evidence for processing in different parallel pathways.

## Differences in redundancy and synergy between tasks

The employed tasks all showed distinct patterns of synergistic and redundant dynamics. The Roving Oddball Task elicited synergistic information mostly within the ERP signal, while the local deviant in the Local/Global Task displayed temporally distributed synergy within both ERP and BB signals. A possible explanation for this is that the MMN-response for the Roving Oddball Task could primarily reflect stimulus-specific adaptation at the level of the auditory cortex[42], while the Local/Global Task shows smaller effects relating to SSA due to the 20-sequence adaptation period at the start of each testing run[28,31].

While the local deviant in the Local/Global Task showed highly distributed synergistic information across brain areas and for both marmosets, the patterns observed for the global deviant were more marmoset-dependent. Strong effects were observed within signals (ERP and BB) and between brain areas (temporo-frontal) for monkey Ji, but monkey Nr exhibited minimal effects within and between all cortical regions (see Fig. S2). A speculative explanation for the lack of a global effect is that higher-order deviants can be driven by top-down attention[43,44]. In this case, the lack of effects in Nr could be simply explained by a lack of interest in the experimental stimuli.

## Distributed processing across cortical areas: implications for predictive coding

Our findings might have ramifications for predictive coding theories. For example, the information encoding PEs was not merely redundant but also highly synergistic across areas. In principle, the lack of redundancy between PEs is inconsistent with hierarchical predictive coding (HPC) theory because HPC entails that prediction errors are independently generated in different levels of the hierarchy[13,45]. However, synergy corresponds to the extra information obtained when signals are considered together, suggesting that there is a more holistic representation of PE rather than just "independently" generated PEs, with the correlational structure of the signals conveying additional information. Possibly, this synergy results from the recurrent (including long-distance feedback) interactions across many nodes, as suggested by the results of the simulations obtained with a brain-constrained neurocomputational model. Thus, these findings suggest that PEs might be encoded by distributed processing rather than local and independent processing.

Overall, our results support the idea that PE information is broadcasted by transient, aperiodic neural activity across the cortex (i.e. ERPs and BB signals)[24]. By dissociating the type of information encoded by these inter-real interactions, we have shown that PEs not merely encode common information but also complementary information between brain signals. Our results demonstrate that distributed representations of prediction error signals across the cortical hierarchy are highly synergistic.

## Methods

### Data acquisition

This study used ECoG recordings from five adult male common marmosets (*Callithrix jacchus*). The details of the datasets for three of the monkeys (Kr, Go and Fr) have been described previously in[26,27], and[28] for two of the monkeys (Ji and Nr).

For marmosets Kr, Go and Fr, (i.e., animals that performed the Roving Oddball Task) the ECoG recordings were acquired in a passive listening condition while the monkeys were awake. During the recording sessions, the monkeys Go and Kr sat on a primate chair in a dimly lit room, while monkey Fr was held in a drawstring pouch, which was stabilized in a dark room. Every session lasted for about 15 min of which the first 3 min of data were used for various standard stimuli and the remaining 12 min of data acquisition were dedicated to the Roving oddball sequences. For the data analysis, we acquired a total of three sessions for monkey Fr, which resulted in 720 (240 × 3) standard and deviant trials, and six sessions for monkeys Go and Kr, resulting in 1440 (240 × 6) standard and deviant trials. For the recordings, a multi-electrode data acquisition system was used (Cerebus Blackrock Microsystems, Salt Lake City, UT, USA) with a band-pass filter of 0.3–500 Hz and then digitized at 1 kHz. In the signal pre-processing, those signals were re-referenced using an average reference montage, and high-pass filtered above 0.5 Hz, using a 6th-order Butterworth filter.

The recording was done with chronically implanted, customized multielectrode ECoG electrode arrays (Cir-Tech Inc., Japan). Before implantation with the ECoG electrode arrays, the monkeys were anesthetized and further suffering was minimized. All electrodes were implanted in epidural space; 28 in the left hemisphere and an additional 4 in the frontal cortex of the right hemisphere of monkey Fr, 64 in the right hemisphere of monkey Go, and 64 in the right hemisphere of monkey Kr. In the 32-electrode array, each electrode contact was 1 mm in diameter and had an inter-electrode distance of 2.5–5.0 mm[26]. In the 64-electrode array, each electrode contact was 0.6 mm in diameter and had an inter-electrode distance of 1.4 mm in a bipolar pair[46]. The electrode arrays covered the temporal, parietal, frontal, and occipital lobes.

For marmosets Ji and Nr, (i.e., the animals that performed the Local/Global Task) the ECoG recordings were also acquired in a passive listening condition while the monkeys were fully awake. The monkeys were seated in sphinx position with their head fixed in a sound-attenuated and electrically shielded room. The recording was done with chronically implanted, multielectrode (96) ECoG electrode arrays (Cir-Tech Inc., Japan). For data analysis, electrodes in the temporal and frontal cortices of the marmosets were used. This was done due to the public availability of the data from these electrodes[28]. Monkey Ji had a total of 27 electrodes (16 temporal, 11 frontal), and monkey Nr had a total of 39 electrodes (25 temporal, 14 frontal). The data was recorded with a Grapevine NIP system (Ripple Neuro, Salt Lake City, UT) with a sampling rate of 1 khz.

All surgical and experimental procedures were performed following the National Institutes of Health Guidelines for the Care and Use of Laboratory Animals and approved by the RIKEN Ethical Committee (No. H26-2-202, for monkeys Kr, Go, and Fr and No. W2020-2-008(2) for monkeys Ji and Nr). The locations of the implanted electrodes of each monkey are found in Fig. 2. Data was collected in MATLAB 2016b (MathWorks Inc., Natick, MA, USA) using the Psychophysics Toolbox extensions.

### Experimental tasks

For the Roving Oddball Task, monkeys Kr, Go and Fr were subjected to a Roving Oddball Task[27]. Trains of 3, 5, or 11 repetitive single tones of twenty different frequencies (250–6727 Hz with intervals of 1/4 octave) were presented in a pseudo-random order. Within each tone train the presented tones had the same frequency, but between tone trains the frequency was different. As the tone trains followed each other continuously, the first tone of a train was considered an unexpected deviant tone, because the preceding tones were of a different frequency, while the expected standard tone was defined as the final tone in a train because the preceding tones were of the same frequency (Fig. 1a). The presented tones were pure sinusoidal tones that lasted for 64 ms (7 ms rise/fall) and the time between stimulus onsets was 503 ms. Stimulus presentation was controlled by MATLAB (MathWorks Inc., Natick, MA, USA) using the Psychophysics Toolbox extensions (Brainard and Vision, 1997). Two audio speakers (Fostex, Japan) were used to present the tones with an average intensity of 60 dB SPL around the animal's ear.

For the Local/Global Task, monkeys Ji and Nr were subjected to a standard Local/Global auditory oddball task[28]. The monkeys heard tone trains with either a local regularity (five identical tones played in a sequence; xxxxx) or global regularity (five tones, the first four of which were identical, and where the fifth was of a different frequency; xxxxY). To create a local deviant, the last tone of the local tone train (xxxxx) was sometimes played at a different frequency as the earlier tones in the train (local deviant; xxxxY). To create a global deviant, the last tone of the global tone train (xxxxY) was sometimes played with the same frequency as the earlier tones in the train (global deviant; xxxxx). The frequencies for the tones x or Y were either 707 or 4000 Hz. The presented tones were pure sinusoidal tones that lasted for 50 ms with an intertone interval of 150 ms, and they were presented to the monkeys bilaterally with two speakers (Fostex, Japan) from the distance of approximately 20 cm from the head with the average intensity of 70 DB.

Each testing period started with a 14-second resting phase, which was followed by a habituation period during which the specified standard (local or global) was presented 20 times to ensure that the monkey learns the regularity of the tone trains. For a testing run, three blocks of 25-tone trains were played, with a 14 s resting phase in between. Out of the 25 trials, 20 (80 percent) were of the specified standard (local or global) and five (20 percent) were deviants. For the global deviants, more than one local standard was always played after to ensure global consistency. Each run lasted for 6 min and 46 s, and each session consisted of 3–4 local standard and 3–4 global standard runs, depending on the marmoset's performance during the day. The order of the tasks was randomised, and the frequencies for tones x and Y were balanced. For the analysis, the number of trials for standard and deviant trials had to be equal. This resulted in 330 (local deviant) and 243 (global deviant) trials for Monkey Ji, 251 (local deviant), and 212 (global deviant) for Monkey Nr.

### ERP and BB analyses

For further analysis, the raw ECoG voltage responses have been transformed into ERP and BB as described in[27]. In brief, common average referencing was used to re-reference the ECoG recordings across all electrodes, and the data was downsampled to 500 Hz. For obtaining ERPs, a low-pass filter of 1–40 Hz was applied for the ERP analysis. Standard and deviant tones were categorized as described before. Epochs of −100 ms to 350 ms around the onset of the tones were taken, and a baseline correction was applied by subtracting the mean voltage during the 100 ms period before the stimulus onset from the total epoch.

To obtain the BB, spectral decoupling of the raw ECoG was carried out[27,29]. In brief, epochs of −100 ms to 350 ms around the onset of the tones were used to calculate discrete samples of power spectral density (PSD). Then, trials from both conditions were grouped and individual PSDs were normalized with an element-wise division by the average power at each frequency, and the obtained values were log-transformed. To identify components of stimulus-related changes in the PSD, a principal component method is applied. This consists of calculating the covariance matrix between the frequencies. The

eigenvectors of this decomposition are called Principal Spectral Components (PSCs), and reveal distinct components of neural processing, hence enabling us to identify stimulus-related changes in the PSD. Afterward, the time series were z-scored per trial to get intuitive units, then exponentiated and subtracted by 1. Finally, a baseline correction was performed by subtracting the mean value of the pre-stimulus period of −100 to 0 ms.

Both for the ERP and BB signals some electrodes were excluded from further analysis. This was done because the signal was absent or clearly erroneous. Electrode 18 in Fr was excluded from the ERP analysis, while electrodes 18 in Fr, 30, 44, 45 in Go, and 30 in Kr were excluded from the BB analysis.

## Mutual information analyses

To quantify the MI between the stimulus class and the ECoG signal (both ERP and BB), the GCMI toolbox (Gaussian Copula Mutual Information)[15] was used. This toolbox calculates the MI based on the Gaussian copula the raw ERP or BB data transforms to. The approach combined a permutation test with 1000 permutations with a method of maximum statistics to correct for multiple comparisons. Using all available trials, the signal at every time point was permuted 1000 times for each electrode, randomly assigning the stimulus class labels each time. The maximum value at each time point was taken, and the 95th percentile of this value was used as the threshold for significance. This method corrects for multiple comparisons and provides a Family-Wise Error Rate (FWER) of 0.05. Electrodes with significant mutual information between standard and deviant trials were selected as electrodes of interest, and the co-I between them was estimated for the ERP and broadband signals separately.

## Co-Information analyses

We quantified co-I within signals (single electrodes) and between signals (between pairs of electrodes) using the GCMI toolbox[15]. The co-I was calculated by comparing signals on a trial-by-trial basis. This resulted in a quantification of the information content, redundant or synergistic, between the two signals. The co-I was calculated in the following way:

$$coI(X;Y;S) = I(X;S) + I(Y;S) - I(X,Y;S) \qquad (1)$$

For each time point, $I(X;S)$ corresponds to the mutual information (MI) between the signal at recording site X and stimuli class S. $I(Y;S)$ corresponds to the MI between the signal at recording site Y and stimuli class S. Finally, $I(X,Y;S)$ corresponds to the MI between stimuli class S combining signals from recording sites X and Y.

For each neural marker of auditory PE (i.e., ERP and BB), co-I was computed for each pair of tones (standard and deviants) within recording sites in A1 and frontal regions (Figs. 3, 5 and Figs. S1–S2), and between A1 and frontal regions (Figs. 4, 6 and Figs. S3–S8). Positive co-I shows that signals between recording sites contain redundant, or overlapping, information about the stimuli. Negative co-I corresponds to the synergy between the two variables: the information when considering the two variables jointly is larger than when considering the variables separately.

Figure 1c shows a schematic representation of co-I between two signals. It shows the independent information that response 1 and response 2 (both in white) contain. If there is an overlap in the information that is being represented by the two signals, there is a redundancy (red color) in the information that the two responses contain. If the two signals considered together contain more information than could be expected based on the information present in the individual signals, there is synergy (blue color). Statistical analyses of co-I charts were performed by using a permutation test with 1000 permutations and using the same maximum statistics method described for the MI analyses, resulting in an FWER of 0.05.

Note that MI and co-I values are reported in units of bits. A value of 1 bit corresponds to a halving of the uncertainty of the trial state when observing the neural response. It is important to keep in mind though that these information values are the average per sample. Here we use a sampling rate of 500 Hz, so a value of 0.01 bits/sample corresponds to an approximate information rate of 5 bits/second.

## Neurocomputational experiments

**Model architecture and function.** To investigate the neural mechanisms underlying the generation of the PE responses observed experimentally in the marmoset brain during oddball presentation of auditory tones we took an existing six-layer-deep, neural network architecture[22,23,35] closely mimicking neuroanatomy and neurophysiology of six perisylvian areas in the left hemisphere of the human brain involved in spoken language and auditory processing, and adapted it for the present study's needs.

This choice was motivated by the observation that, like other non-human primates, marmosets are known to be highly vocal and exhibit active vocal communication among conspecifics[47–49]; furthermore, the existing architecture has been previously used to simulate and explain well-documented neurophysiological patterns of event-related potentials observed during language processing and oddball stimulation with familiar and unfamiliar sounds[23,32]. The model closely reflects the functional and structural features of the mammalian cortex, and incorporates the following neurobiological and neurophysiological constraints:

1.  Six cortical areas are modelled, three in the superior temporal and three in the inferior frontal lobes, constituting the marmoset homologues of Brodmann Areas (BAs) 41 (labelled A1 in Fig. 1f), 42 (labelled AB), and 22 (labelled PB) in the superior temporal gyrus, and of BAs 44 and 45 (labelled PF), 6V (PM), and 4 (M1) in the inferior frontal gyrus in humans;
2.  Between-area links in the model (green, purple, and black arrows in Fig. 1f) reflect known neuroanatomical links between corresponding brain areas in the marmoset (see next section below); recurrent (within-area) connections (golden arrows) are also modelled, in line with known properties of the mammalian cortex[50,51];
3.  Between- and within-area links do not implement *all-to-all* connectivity between cells, but sparse, patchy, and topographic projections, with synaptic links established probabilistically (the probability of two cells being connected decreasing with the distance; see[51–53] and initialised to weak and random efficacy values;
4.  Local lateral inhibition[54,55] and area-specific global regulation mechanisms (referred to as local and global inhibition, respectively)[51,55,56];
5.  Single cells' neurophysiological dynamics, including sigmoid transformation of membrane potentials into neuronal outputs, as well as adaptation and temporal summation of inputs[57];
6.  Constant presence of uniform uncorrelated white noise (simulating spontaneous baseline neuronal firing) in all model neurons[58].

A first difference from the human language cortex is that the location of the marmoset homologue of BA 44—one of the major components of Broca's area[59]—still has not been definitively agreed upon[47]. However, area 6Vb in the marmoset—which, like in man and macaque, is just caudal to 45—exhibits cytoarchitectonic features (a scattered, agranular layer 4) that make it a potential candidate for the BA 44 homologue[47]. In addition, area 6Vb shows a pattern of neuroanatomical connectivity different from that of its dorsal (and more caudal) counterpart 6Va, a premotor area[60]. In previous "human" versions of the architecture, area PF (modelling prefrontal cortex) was defined as including mainly BA45 (and 46v), whereas BA 44 was

subsumed by model area PM. Given the above, and the fact that BA 44 is generally considered a prefrontal cortex area, here we decided to treat Marmoset's area 6Vb as the homologue of the insofar missing BA 44, and to label *both* 45 and 6Vb as model area PF (hence limiting PM to include just area 6Va, homologue of BA 6V).

Structurally, each model area consists of two neuronal layers, one of excitatory and one of inhibitory cells, each containing 625 ($25 \times 25$) cells (see Fig. 1g). Functionally, model cells are graded-response units, each representing a cluster of excitatory pyramidal cells or inhibitory interneurons. The specifics of the computational implementation (including the within-area structure and single-cell functional features) are analogous to those implemented in previously published versions of the architecture (for details, see[22,32] and can be found in the Supplementary Methods for completeness.

A second crucial aspect that distinguishes human, macaque, and marmoset brains is the structural connectivity between the relevant homologue areas. Our present approach, which builds upon and is in line with several previous studies carried out with this neurocomputational architecture[22,23,32,33,35–37,61], is to implement a fully brain constrained model. More precisely, we impose that, for any two model areas, synaptic projections between them are realised only if experimental evidence indicates the presence of neuroanatomical links between the two corresponding cortical areas in the marmoset brain. In the following section, we provide such evidence and the rationale based on which the present network architecture (shown in Fig. 1f) was adopted.

**Connectivity of the simulated brain areas.** The implemented model areas can be thought of as grouped into two sub-systems (frontal and temporal), each simulating a hierarchy of three cortical areas consisting of a primary cortex (motor and auditory, respectively), the adjacent higher secondary, and associative multimodal regions. Neuroanatomical studies in the mammalian brain indicate that adjacent cortical areas tend to be reciprocally connected[62,63]. We implemented such next-neighbor connections (black arrows in the network architecture shown in Fig. 1f) in each of the two subsystems based on known evidence from nonhuman primates (including marmosets): within the frontal/motor (PF-PM-M1)[60,62] and within the temporal/auditory (A1-AB-PB)[64–69]. The links connecting the parabelt (area PB) with prefrontal cortex (PF), shown in purple in Fig. 1f, are also realised in line with evidence on known long-distance cortico-cortical white matter fibres in the monkey (arcuate fascicle and extreme capsule) connecting posterior-lateral parts of the temporal cortex (area PB) and inferior prefrontal cortex (area PF)[60,70–74]. Finally, although less strong and richly developed than in humans[75,76], the presence of higher-order "jumping" connections between non-adjacent areas in the model (green arrows in Fig. 1f) has been documented also in monkeys (including in marmosets). Specifically, neuroanatomical studies indicate that A1 is directly connected to PB[62,67–69,77], that AB is connected to PF[64,78–81], that PB and PM are linked[73,82] and that PF—here including areas 45 and 6Vb—is also directly connected to M1[83,84].

A previous modelling study using a neurocomputational architecture analogous to the present one looked at the effects of qualitative and quantitative differences between monkey's and human's perisylvian areas connectivity on verbal working memory[35]. In that study, the architecture used to simulate the monkey brain did not implement any of the jumping links that we included in the present model of the marmoset's cortex. The authors, however, did acknowledge that the extant evidence does not imply a "*complete absence of jumping links in nonhuman primates*". In addition, none of the neuroanatomical studies used to constrain that modelling work included data from marmoset monkeys (the evidence about non-human primate connectivity used relied on macaques or chimpanzees). Finally, as Schomers and colleagues clarified, their work focused on modelling the major structural connectivity differences between monkeys and

humans rather than on modelling the full complexity of the neuroanatomical connections of either species[35]. Hence, given that - albeit weaker and less developed than in humans - jumping links in non-human primates (including in the marmoset) do appear to exist, their study and the present one should not be considered as in conflict, but simply as using the same network architecture to address different computational questions.

**Procedures.** To simulate the Roving Oddball Task of Experiment 1, the network was repeatedly presented with stimulus patterns to its auditory cortex (area A1). A stimulus pattern (simulating an auditory tone) consisted of a pre-determined set of 31 cells chosen at random amongst the 25-by-25 cells of area A1 (about 5% of cells). We used 12 different randomly generated stimulus patterns; presenting a stimulus involved activating the 31 cells of the chosen pattern in A1. A single trial consisted of a baseline (ten simulation time-steps long) with no input, followed by 20 time-steps of stimulus presentation, and 20 time-steps of inter-trial interval (no input); stimulus onset asynchronicity was therefore 50 simulation time-steps. A roving paradigm was used, in which 89% of standard (STD) trials were intermixed with 11% of deviant (DEV) trials. A new DEV trial was always preceded by 6-to-10 identical STD stimuli; the new DEV stimulus was chosen at random. The network's output (firing rates of all cells of the six areas) was recorded from the start of the last STD trial to the end of the critical (DEV) trial following it. For each simulation run (a model correlate of a monkey recording) we collected a total of 50 STD and 50 DEV trials. MI and co-I analyses on the simulated data, as well as the statistical contrasts between STD and DEV tones, were performed exactly as described for the in-vivo data.

**Multivariate co-information method (MVCo-I).** Mutual information quantifies statistical dependence on the meaningful effect size of bits. Crucially, these values are additive when combining independent representations. This allows us to quantify representational interactions between electrodes as synergistic or redundant using co-I as described in the manuscript. However, estimating mutual information on high-dimensional responses is challenging. Multi-Variate Pattern Analysis (MVPA) is an approach that has been widely adopted in neuroimaging and neuroscience to deal with high-dimensional signals[85]. MVPA uses techniques from the field of machine learning: namely, supervised learning algorithms and cross-validation, to learn informative patterns in high-dimensional data, and evaluate their generalisation performance (i.e. how well the model could predict in new data). Here we use linear-discriminant analysis to learn the most informative linear combination of channel activity in each region to predict the binary class of the stimulus (i.e. deviant vs standard). There are various metrics for evaluating the cross-validated predictive performance of classification algorithms, for example, overall accuracy, or measures like Area under the ROC curve[86]. These metrics can be used to rank models based on different features (i.e. compare the amount of information in temporal vs frontal regions).

Here, we combine MVPA with information-theoretic co-I to quantify the representational interactions in predictions made from cross-validated multivariate models. To do this, we first apply MVPA in the typical way (here using MVPALight toolbox;[87]) using 10-fold cross-validation (CV). In a 10-fold CV, the overall dataset is randomly separated into 10 disjoint subsets. Then, a model is fit on 9 of those subsets and tested on the 10th, and this is repeated for each of the 10 subsets. Here, we take the decision value of the learned classifier (the value of the linear combination of the weights and the data, which would then be thresholded to make the classification) for each test set trial. This quantifies how strongly the informative pattern the classifier had learned was present in the data on that trial. We combine the test-set decision values from all 10 different CV repetitions and calculate the mutual information between these out-of-sample decision values and

the true stimulus value on each trial[88,89]. We have used MVPA to reduce the activity of the multi-channel region into a single scalar value: the decision value (d-val).

We can repeat the MVPA analysis for each time point of the stimulus-locked epochs. Often, temporal cross-decoding[85] is employed with MVPA to compare the consistency of the informative patterns over time. For this method, a classifier is trained at time $t$, and then tested (in the hold-out test folds) at other times. If it can decode, it shows the same pattern that was learned at time $t$, which is information at other times. However, this can only compare between data sets or conditions that are in the same physical space: i.e. we can cross-decode across time within one brain region, but we cannot compare between two different brain regions, because there is no way to apply the linear weights learned in the frontal region to the completely different temporal electrodes. Combining MVPA with co-Information (MVCo-I) overcomes this limitation. We compute co-I between the cross-validated decision values of different classifiers. This admits the same interpretation as for the channel-wise analysis. Redundancy shows that there is common information accessed by the two decoding models. Synergy means that there is a super-additive boost in the information available when considering the pattern activation from both models together. When estimating the joint information for the co-information calculation we take the maximum of the individual region MI (because the data processing inequality tells us this is a lower bound on the information that can be extracted from the joint response), the MI from the combined d-vals (2D signal; this has the advantage of being a low dimensional response for MI calculation, and being the optimal informative signal from each region) and the MI from a joint MVPA model fit to the combination of channels from both regions (1D d-vals, but which has the possibility to include synergistic information between the regions which we want to capture with this measure) (Fig. S11).

We apply this methodology here in two ways. First, we look at within-area MVCo-I. For this, we train CV classifier models separately at each time point. We then calculate the co-information between two time points using the cross-validated decision values of the two models. Note, that a crucial difference between this and the temporal cross-decoding method is that we always use the model that is learned to optimally decode information at that time point. Cross-decoding can tell if the same pattern is informative, but we can see redundant information even when the informative pattern changes. We can then compute MVCo-I between regions in the same way.

Data was analyzed using MATLAB 2019b (MathWorks Inc., Natick, MA, USA); and the open-source Gaussian Copula Mutual Information (GCMI) Toolbox[15].

### Reporting summary

Further information on research design is available in the Nature Portfolio Reporting Summary linked to this article.

## Data availability

The original raw data of marmosets Fr, Kr, and Go used in this study is publicly available at: http://www.www.neurotycho.org/auditory-oddball-task. The original raw data of marmosets Ji and Nr used in this study is publicly available at: https://datadryad.org/stash/dataset/doi:10.5061/dryad.j3tx95xfp Source data are provided with this paper.

## Code availability

The MATLAB/Python toolbox GCMI (Gaussian Copula Mutual Information) used in this study is publicly available at: https://github.com/robince/gcmi.

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

## Acknowledgements

ACJ thanks Dr. Daniel Bor, Dr. Valdas Noreika, Dr. Tristan Bekinschtein, and Dr. Dora Hermes for contributing to valuable discussions. MG thanks Dr. Friedemann Pulvermüller for the invaluable discussions and Dr. Thomas Wennekers for his substantial help in developing the original neurocomputational architecture; special thanks also to Eamonn Martin and the Computing Department at Goldsmiths for providing the infrastructure and support needed to run the neural simulations. ACJ is supported by a Bial Foundation Grant (ID: A-29477) and an ANID/FONDECYT Regular (1240899) research grant. MV is supported by an ERC Starting Grant [SPATEMP, EU], a BMBF (Germany) Grant [Computational Life Sciences, project BINDA, 031L0167], DFG VI Grants (908/5-1 and 908/7-1), the NWO VIDI, and the Dutch Brain Interface Initiative. MK financially supported this project with two AMED Brain/MINDS Grants (JP19dm0207001 and JP19dm0207069). KJM supported this project via the Foundation for OCD Research and NIH U01-NS128612.

## Author contributions

Conceptualization: ACJ, RI, and MV. Data analysis: FG, ACJ, LR and JÄ. Visualization: FG, ACJ, and JÄ. Marmoset recordings and surgeries: MK. Software and methods for marmoset- and simulation-data analysis: RI, MJ, LR, and KJM. Neurocomputational modelling: MG. Writing and editing: ACJ, FG, JÄ, CU, MG and MV. Supervision: ACJ.

## Competing interests

The authors declare no competing interests.
