## [Peer Review File · Nature Communications]

Distributed representations of prediction error signals across the cortical hierarchy are synergisticREVIEWER COMMENTS

Reviewer #1 (Remarks to the Author):

The manuscript addresses the question of information processing in distributed neural networks and specifically, whether local activities in distinct brain regions are redundant or synergistic. The notion of synergy is defined here as evidence of a form of information sharing (called here co-information) between the neural activity patterns captured at distinct recording sites. These ideas are tested using a remarkable set of widespread electrocorticographic (ECoG) recordings in 5 common marmosets, 3 and 2 of them subjected to two different auditory tasks known to elicit prediction error signals. The main message of the report is that prediction error signals are encoded across multiple brain sites in a synergistic manner, in addition to well-documented regional manifestations in e.g., auditory regions.

Although the scientific question is significant and the nature of the data is outstanding, I find the premises of the study and the approaches rather unclear, and the actual effects in the data unconvincing.

A major issue with the interpretation of redundant vs. synergistic effects between pairs of recording sites is with the lack of control for possible trivially redundant information contributed by a third site or more. Indeed, I understand that synergistic vs. redundant effects are ascertained on the basis of pairwise measures of co-information between two recording sites. These pairwise measures rest on the unstated hypothesis that network interactions in the marmoset brain take place between two brain nodes, irrespective of the activity taking place at a third location, or more. This issue of partial measures of interactions, is well known in network neuroscience but seems to have been overlooked by the authors unfortunately. It is a significant issue because effects presently interpreted as synergistic between two recording sites, may actually be caused by the activity at a third site or more, and be redundant, not synergistic with the activity these sites. Therefore, the measures of co-information between two sites need to be assessed conditionally to the activity at the other recording sites, just like when partial correlations are derived from fMRI or other time series to establish causal connexions between brain regions. Although controlling partial redundancy may not be crucial in most network studies, it is crucial here because the entire purpose of the study is to establish whether pairwise brain site interactions are redundant or not. To do this properly, it is essential to control for possible redundant interactions from regions outside the tested pair of brain sites.

Another, more pragmatic issue with the paper is the lack of clarity in the presentation of the core results, and essentially the main body of results from Figure 3 onwards. A large number of temporal co-information diagrams are stacked in these figures with relatively poor guidance for visualization (some colorbars are missing) and with statistically significant effects of an uncertain spatio-temporal structure.

The data from the computational model of brain networks is also only casually (qualitatively) compared to the empirical electrophysiological data, and related statements such as "[the model] accurately replicates critical neurobiological and neuroanatomical features of the mammalian cortex", "we immediately observed..." etc. are clearly exaggerated.

Overall, although the manuscript addresses a significant question with rare data, I find there are severe shortcomings in the approach, the presentation and the interpretation of the results.

Reviewer #2 (Remarks to the Author):

This paper analyzes synergy between different cortical regions and time delays for conveying prediction error signals. The authors report that interactions are primarily synergistic. The analysis is based on event-related potentials (ERP) and broad band (BB) signals. Overall, the results have the potential to provide a useful contribution to the field. However, the observed size of synergy/redundancy is very small, on the order of ~ 0.01 . Importantly, the mutual information values have not been corrected for finite data bias. One approach for correcting for this bias is described here: <https://pubmed.ncbi.nlm.nih.gov/10935917/>. Once the values are corrected for the bias, it is likely that none of the results will be statistically significant.

Please note that the error-bars were not described in figure legends and it is not clear what the shaded regions represent, such as whether these are standard errors of the mean, standard deviation, or 95% confidence intervals.

The authors should acknowledge and link better to a larger body of work that considered co-information in computational systems. In particular, <https://www.jneurosci.org/content/23/37/11539> provided detailed discussion of co-information and how it should be interpreted. Regarding predictive information, the work by Israel Nelken should be cited and discussed, e.g. <https://pubmed.ncbi.nlm.nih.gov/22248537/> and <https://www.nature.com/articles/nn1032>

Reviewer #3 (Remarks to the Author):

In this exciting paper, Gelens et. al. use two auditory protocols in non-human primates with ECOG to study how prediction error is processed across brain regions. To that end, they computed co-information (a method that distinguishes redundant from synergic information processing). This paper is fairly written; the authors present an impressive number of results that are, at certain points are difficult to integrate for the reader. Overall, I am impressed with the work done by the authors. This work will certainly constitute an important addition to the field. However before recommending the publication, there are some aspects that the authors need to clarify to validate their results and the conclusions entirely.

1) My main concern is that in all of the presented analyses, there is a bias towards synergic interactions that the processing of the stimulus itself might not drive it, but the interaction with unrelated ongoing activity. This can be directly observed in the interactions between the baseline period (-100 to 0 ms) and the post-stimulus period (0 to 300 ms) in all cases (experimental data - Figures 3, 4, 5, 6 – and model - Figures 7 and 8-) there is a synergic interaction (blue). In my interpretation, this synergic interaction between stimulus-related and non-related activity is a bias that probably affects the rest of the analysis and needs to be corrected. A potential way to do it is to do a baseline correction of the co-information computation and remove the mean value of these periods from the rest of the analysis. This will probably reduce the quantified synergy (and augment the redundancy). The authors must quantify if the results (and the conclusions) change after this control.

2) The authors present a computational model to explore potential explanations for the results. The authors argue that comparing a computational model including feedforward and feedback connections and one with only feedforward alone demonstrates that feedback connections are responsible for the emergence of synergic processing. I'm afraid I have to disagree with the authors that the models themselves can demonstrate any neuronal mechanism. Models are helpful to interpret data and generate predictions that can be subsequently experimentally tested. The results obtained with the model are not evidence of how the mechanism is implemented in the brain. In addition, the results presented by authors comparing the two types of models (Figures 7 and 8) are suggestive of a reduction of synergy. Still, there's no quantification of this result (overall, the simulations coming out of the two models look quite similar -panels B/D and C/F in figure 7 and panels B/D in figure 8 -).

Reviewer #1 (Remarks to the Author):

The manuscript addresses the question of information processing in distributed neural networks and specifically, whether local activities in distinct brain regions are redundant or synergistic. The notion of synergy is defined here as evidence of a form of information sharing (called here co-information) between the neural activity patterns captured at distinct recording sites. These ideas are tested using a remarkable set of widespread electrocorticographic (ECoG) recordings in 5 common marmosets, 3 and 2 of them subjected to two different auditory tasks known to elicit prediction error signals. The main message of the report is that prediction error signals are encoded across multiple brain sites in a synergistic manner, in addition to well-documented regional manifestations in e.g., auditory regions.

Response: We thank the reviewer for their initial evaluation and constructive criticism, and we appreciate the opportunity to improve our methodological approach and the overall quality of the manuscript. We have added new analyses and figures, as well as discussion, to address the reviewer's concerns point by point:

Although the scientific question is significant and the nature of the data is outstanding, I find the premises of the study and the approaches rather unclear, and the actual effects in the data unconvincing.

A major issue with the interpretation of redundant vs. synergistic effects between pairs of recording sites is with the lack of control for possible trivially redundant information contributed by a third site or more. Indeed, I understand that synergistic vs. redundant effects are ascertained on the basis of pairwise measures of co-information between two recording sites. These pairwise measures rest on the unstated hypothesis that network interactions in the marmoset brain take place between two brain nodes, irrespective of the activity taking place at a third location, or more. This issue of partial measures of interactions, is well known in network neuroscience but seems to have been overlooked by the authors unfortunately.

Response: Thanks for the comment. We would like to emphasise however that typical network studies relate the activity in different areas. Such activity-based analyses typically consider a correlation (or mutual information) between the time-varying activity of two regions. This correlation might reflect a genuine connection, but could also reflect a third region that independently connects to both regions. Conditional measures (e.g. partial correlation, conditional mutual information) can address this (with some caveats – for example, conditioning out does not actually fully remove the contribution of the conditioned node, because conditioned measures include synergistic effects, e.g. Williams and Beer, 2010). However, applying these sorts of measures can be challenging as one must condition out all other recorded nodes. In any case, there may always be other nodes, which for whatever practical reasons were not accessed with the experimental setup (see Mehler & Kording, 2018).

While we acknowledge the limitation that our study is based on pairwise representational interactions, we believe the focus on representational interactions of experimentally controlled prediction errors means this is a novel and valuable contribution. We would also like to emphasise that our primary scientific goal concerns two inherently pairwise questions. First, relating the representation of stimuli between different time points (c.f. The temporal generalization cross-decoding method, see King & Dehaene, 2014), and second, relating activity between auditory sensory processing regions, and frontal regions often implicated in control or higher level processing. These are pairwise question and so we argue pairwise measures of representational interaction are sufficient and useful to approach them. To address the reviewer's comments about the exclusion of certain sites in the channel-wise analysis, we have extended the manuscript with a new methodology that uses a multivariate pattern analysis of each region. We still perform a pairwise comparison of the representation between frontal and temporal regions, but now we can include all channels within each region. We detail this below.

In order to directly address the reviewer's concern, we have developed an entirely new approach for computing co-information based on the response across multiple electrodes (coined as MVCo-I Method). In particular, we have applied a multivariate analysis approach that uses machine learning to capture the best linear representation of the prediction error signal across a whole region, and we have repeated our co-information analyses within and between the two brain regions of interest using the classifiers' outputs (frontal and temporal). As the reviewer might notice in the next series of new control results, the patterns of synergy and redundancy observed both within and between regions for both tasks replicate the results observed using the electrode pair-wise comparisons. The description of the MVCo-I method, as well all as the changes made in the manuscript regarding its usage as a control analysis for potential informative spatial patterns of redundancy and synergy that might be missed from our original electrode pairwise co-I analyses.

Lines 254-283: *“Although the per-electrode and electrode-pair analyses of synergy and redundancy exploit the optimal spatial resolution of the recording modality across temporal and frontal regions, they could also miss information encoded in the spatial pattern both within and between temporal and frontal areas. They could therefore potentially miss synergy or redundancy that is only apparent when considering multiple electrodes together, either due to low signal to noise ratio within each channel, or because of a genuinely distributed informative spatial pattern. This might be particularly relevant for the ERP signals that showed extensive temporal and frontal PE effects (Figure 1A,B). Thus, to account for potential informative spatial patterns of redundancy and synergy in ERP responses, and to reduce any concern about high-order interactions between channels within each region in the pairwise channel analysis, we complemented our analyses by computing co-information based on the response across multiple electrodes (MVCo-I: Multi-Variate Co-information) (Figure S11 and S12). In brief, we have applied a cross-validated multivariate analysis approach that uses machine learning to capture the best linear representation of the prediction error signal across a whole region, and we have repeated our co-information analyses within and between the two brain regions of interest using the classifiers' outputs (frontal and temporal) (see Methods).*

The MVCo-I analyses within-region (Figures S13 and S15) and between-regions (Figures S14 and S16) showed comparable co-I in terms of synergistic and redundant dynamics observed in the per-electrode (Figures 3,5) and in the between-electrodes (Figures 4,6) analyses, but with increased statistical power (i.e., increased MI).”

Lines 993-1078:

“Multivariate Co-Information Method (MVCo-I):

Mutual information quantifies statistical dependence on the meaningful effect size of bits. Crucially, these values are additive when combining independent representations. This allows us to quantify representational interactions between electrodes as synergistic or redundant using co-I as described in the manuscript. However, estimating mutual information on high-dimensional responses is challenging. Multi-Variate Pattern Analysis (MVPA) is an approach that has been widely adopted in neuroimaging and neuroscience to deal with high-dimensional signals (King and Dehaene, 2014). MVPA uses techniques from the field of machine learning: namely, supervised learning algorithms and cross-validation, to learn informative patterns in high-dimensional data, and evaluate their generalisation performance (i.e. how well the model could predict in new data). Here we use linear-discriminant analysis to learn the most informative linear combination of channel activity in each region to predict the binary class of the stimulus (i.e. deviant vs standard). There are various metrics for evaluating the cross-validated predictive performance of classification algorithms, for example, overall accuracy, or measures like Area under the ROC curve (Poldrack et al., 2020). These metrics can be used to rank models based on different features (i.e. compare the amount of information in temporal vs frontal regions). Here, we combine MVPA with information-theoretic co-information to quantify the representational interactions in predictions made from cross-validated multivariate models. To do this, we first apply MVPA in the typical way (here using MVPALight toolbox; Treder, 2020) using 10-fold cross-validation (CV). In a 10-fold CV, the overall

dataset is randomly separated into 10 disjoint subsets. Then, a model is fit on 9 of those subsets and tested on the 10th, and this is repeated for each of the 10 subsets. Here, we take the decision value of the learned classifier (the value of the linear combination of the weights and the data, which would then be thresholded to make the classification) for each test set trial. This quantifies how strongly the informative pattern the classifier had learned was present in the data on that trial. We combine the test-set decision values from all 10 different CV repetitions and calculate the mutual information between these out-of-sample decision values and the true stimulus value on each trial (Yan et al., 2023; Yan et al; 2023b). We have used MVPA to reduce the activity of the multi-channel region into a single scalar value: the decision value.

We can repeat the MVPA analysis for each time point of the stimulus-locked epochs. Often, temporal cross-decoding (King and Dehaene, 2014) is employed with MVPA to compare the consistency of the informative patterns over time. For this method, a classifier is trained at time t , and then tested (in the hold-out test folds) at other times. If it can decode, it shows the same pattern that was learned at time t , which is information at other times. However, this can only compare between data sets or conditions that are in the same physical space: i.e. we can cross-decode across time within one brain region, but we cannot compare between two different brain regions, because there is no way to apply the linear weights learned in the frontal region to the completely different temporal electrodes. Combining MVPA with co-Information (MVCo-I) overcomes this limitation. We compute co-information between the cross-validated decision values of different classifiers. This admits the same interpretation as for the channel-wise analysis. Redundancy shows that there is common information accessed by the two decoding models. Synergy means that there is a super-additive boost in the information available when considering the pattern activation from both models together. When estimating the joint information for the co-information calculation we take the maximum of the individual region MI's (because the data processing inequality tells us this is a lower bound on the information that can be extracted from the joint response), the MI from the combined dvals (2D signal; this has the advantage of being low dimensional response for MI calculation, and being the optimal informative signal from each region) and the MI from a joint MVPA model fit to the combination of channels from both regions (1D d-vals, but which has the possibility to include synergistic information between the regions which we want to capture with this measure).

We apply this methodology here in two ways. First, we look at within-area MVCo-I. For this, we train CV classifier models separately at each time point. We then calculate the co-information between two-time points using the cross-validated decision values of the two models. Note, that a crucial difference between this and the temporal cross-decoding method is that we always use the model that is learned to optimally decode information at that time point. Cross-decoding can tell if the same pattern is informative, but we can see redundant information even when the informative pattern changes. We can then compute MVCO-I between regions in the same way."

In this method, the classifier can extract all the (linearly available) information in each region. We hope this addresses some of the reviewer's concerns about redundant third channels, at least in the regions we consider.

References:

- Poldrack, R. A., Huckins, G., & Varoquaux, G. (2020). Establishment of best practices for evidence for prediction: a review. *JAMA psychiatry*, 77(5), 534-540.
- Treder, M. S. (2020). MVPA-light: a classification and regression toolbox for multi-dimensional data. *Frontiers in Neuroscience*, 14, 289.
- King, J. R., & Dehaene, S. (2014). Characterizing the dynamics of mental representations: the temporal generalization method. *Trends in cognitive sciences*, 18(4), 203-210.
- Yan, Y., Zhan, J., Ince, R. A., & Schyns, P. G. (2023). Network communications flexibly predict visual contents that enhance representations for faster visual categorization. *Journal of Neuroscience*, 43(29), 5391-5405.
- Yan, Y., Zhan, J., Garrod, O., Cui, X., Ince, R. A., & Schyns, P. G. (2023). Strength of predicted information content in the brain biases decision behavior. *Current Biology* (in press).

MVCo-I: Multivariate Co-Information

Figure S11. Schematic of the MVCo-I method. To calculate co-information between multivariate responses we use 10-fold cross-validation (CV) with Linear Discriminant Analysis (LDA). For each of the 10 hold-out folds, we train an LDA classifier to discriminate the stimulus category of a trial (i.e. deviant vs standard tone). We then compute the classifier decision values (d-vals; i.e. the linear combination of the learned pattern weights and the raw data) on each trial for the hold-out fold. We then concatenate the CV d-vals from all folds. We have used LDA as a cross-validated supervised dimensionality reduction method to obtain a one-dimensional representation of the region's activity at that time point, which is maximally discriminative between conditions. We can then compute the co-information between the ground-truth stimulus class of each trial, and the CV d-vals from two different classifiers (i.e. either from the same region at different time points, within-area co-I, or from different regions, between-areas co-I). This calculation is the same as for channel-wise analysis (see Methods).

Anatomy-based electrode selection

Figure S12. Anatomy-based electrode selection for the MVCo-I Method per marmoset. Roving Oddball Task (3 marmosets: Kr, Go and Fr) and Local/Global Task (2 marmosets: Ji and Nr).

Figure S13. Temporal synergy and redundancy within ERP signals in the auditory and frontal electrodes using the MVCo-I Method (Experiment 1: Roving Oddball Task). MVCo-I revealed synergistic and redundant temporal patterns within Temporal ERP (Panel A) and Frontal ERP (Panel B) signals in the auditory cortex. MI (solid traces) between standard and deviant trials for auditory (pink color) and frontal (orange color) responses averaged across the three monkeys. The corresponding electrodes used for the MVCo-I method are depicted in **Figure 1B**. Error bars represent standard error of the mean (S.E.M). Temporal co-I was computed within the corresponding signal (ERP) across time points between -100 to 350 ms after tone presentation. The average of the corresponding electrodes across monkeys is shown for the complete co-I chart (red and blue plots); for positive co-I values (redundancy only; red panel); and for negative co-I values (synergy only; blue plot).

Figure S14. Spatio-temporal synergy and redundancy between auditory and frontal ERP signals using the MVCo-I Method (Experiment 1: Roving Oddball Task). **(A)** MVCo-I revealed synergistic and redundant temporal patterns between temporal and frontal signals. MI (solid traces) between standard and deviant trials for auditory (pink color) and frontal (orange color) responses averaged across the three monkeys. The corresponding electrodes used for the MVCo-I method are depicted in **Figure 1B**. Error bars represent standard error of the mean (S.E.M). Temporal co-I was computed within the corresponding signal (ERP) across time points between -100 to 350 ms after tone presentation. The average of the corresponding electrodes across monkeys is shown for the complete co-I chart (red and blue plots); for positive co-I values (redundancy only; red panel); and for negative co-I values (synergy only; blue plot).

Figure S15. Temporal synergy and redundancy within ERP signals in the auditory and frontal electrodes using the MVCo-I Method (Experiment 2: Local/Global Task). MVCo-I revealed synergistic and redundant temporal patterns within auditory ERP signals in the Local (Panel A) and Global (Panel B) contrasts; and within frontal ERP signals in the Local (Panel C) and Global (Panel D) contrasts. MI (solid traces) between standard and deviant trials for auditory (pink color) and frontal (orange color) responses averaged across the two monkeys. The corresponding electrodes used for the MVCo-I method are depicted in **Figure 1B**. Error bars represent standard error of the mean (S.E.M). Temporal co-I was computed within the corresponding signal (ERP) across time points between -100 to 350 ms after tone presentation. The average of the corresponding electrodes across monkeys is shown for the complete co-I chart (red and blue plots); for positive co-I values (redundancy only; red panel); and for negative co-I values (synergy only; blue plot).

Figure S16. Spatio-temporal synergy and redundancy between temporal and frontal ERP signals using the MVCo-I Method (Experiment 2: Local/Global Task). MVCo-I revealed synergistic and redundant temporal patterns between auditory and frontal ERP signals in the Local (Panel A) and Global (Panel B) contrasts. MI (solid traces) between standard and deviant trials for auditory (pink color) and frontal (orange color) responses averaged across the two monkeys. The corresponding electrodes used for the MVCo-I method are depicted in **Figure 1B**. Error bars represent standard error of the mean (S.E.M). Temporal co-I was computed within the corresponding signal (ERP) across time points between -100 to 350 ms after tone presentation. The average of the corresponding electrodes across monkeys is shown for the complete co-I chart (red and blue plots); for positive co-I values (redundancy only; red panel); and for negative co-I values (synergy only; blue plot).

References:

Williams, P. L., & Beer, R. D. (2010). Nonnegative decomposition of multivariate information. arXiv:1004.2515.
 Mehler, D. M. A., & Kording, K. P. (2018). The lure of misleading causal statements in functional connectivity research. arXiv:1812.03363.

It is a significant issue because effects presently interpreted as synergistic between two recording sites, may actually be caused by the activity at a third site or more, and be redundant, not synergistic with the activity these sites.

Response: We don't agree with this assessment. A major feature of our study is that we do not relate neural activity between two nodes (as in classical network analyses as mentioned above), but instead, we anchor our measures on the experimental stimulus manipulation – i.e. we look at and relate the information different regions carry about the stimulus. Our synergistic interactions are not between the unconstrained resting-state activity of 3 brain regions, but are about an interaction in the representation of an experimentally controlled stimulus contrast (here the oddball PE manipulation) between two regions. This experimental control puts us in the regime of “randomised control trial”, the gold standard for determining causal relations in experimental science, as opposed to the “observational study” analytic regime which is typical for activity-based resting-state network neuroscience.

Given this fundamental difference, we don't agree that the restriction to pairwise representational interaction is such a limitation. By definition, the existence of synergistic information *about* the oddball means there is information that is not available from either region individually but is available from the two together. Either region could be correlated to any degree with any number of third regions but that wouldn't change the fact about the synergy – that the relationship in the activity between these two regions conveys information that is not available from either one alone. In our view, this is a novel and useful statement about distributed neural activity during prediction error. For example, there could be a third site, which has a representation that is redundant with the information that is represented synergistically between the two sites – but this does not affect the synergistic relationship between the two sites. There could be a third site, which conveys information redundantly with either or both of the considered sites, but again, this would change the fact that there is synergy between those two. There could be a third site with activity unrelated to the stimulus, or indeed any other source of common noise or global state to the two regions that could also result in synergy (because observing one gives an estimate of the common noise that can improve the prediction of the stimulus based on the second). But this is just a fact of these sorts of measures, observing information theoretic synergy is a statistical result of the system, not a mechanistic one. So we do not make any resulting claims about connectivity between areas that are the goal of typical resting-state functional connectivity where partially out other areas is of course crucial for such interpretations.

Therefore, the measures of co-information between two sites need to be assessed conditionally to the activity at the other recording sites, just like when partial correlations are derived from fMRI or other time series to establish causal connections between brain regions.

Response: As above, we argue there is a fundamental difference in the experimental goals, the interpretation of the results, and the methodology used in our study compared to typical resting-state, activity-based functional correlations.

Although controlling partial redundancy may not be crucial in most network studies, it is crucial here because the entire purpose of the study is to establish whether pairwise brain site interactions are redundant or not. To do this properly, it is essential to control for possible redundant interactions from regions outside the tested pair of brain sites.

Response: We don't agree with this point. Our scientific question is fundamentally pairwise – we wish to relate prediction error between frontal and temporal regions to gain insight into the hierarchical implementation of this important process. Because our representational interactions are anchored on the experimentally controlled stimulus, we do not see how any putative third region affects our pairwise interpretation. If there was a third subcortical region we could not access which was redundant with activity in both frontal and temporal – this does not affect our conclusions about the redundancy between this pair of regions in any way. It is still the case that the prediction error decoded from frontal regions is redundant with the prediction error decoded from temporal regions. Observing redundancy tells us that there is the same trial-by-trial predictive content in both areas. If this is the case, whether that same information is available in other parts of the brain is an orthogonal question. In this way, the shift from activity correlation to information-theoretic representational interaction changes the interpretation and reduces the concern about third regions.

Nevertheless, as shown at the beginning of the response, to address this concern we have applied a multivariate analysis approach that uses machine learning to capture the best linear representation of the prediction error signal across a whole region, and we repeat our analysis between the two brain regions of interest (frontal and temporal).

Another, more pragmatic issue with the paper is the lack of clarity in the presentation of the core results, and essentially the main body of results from Figure 3 onwards. A large number of temporal co-information diagrams are stacked in these figures with relatively poor guidance for visualization (some colorbars are missing) and with statistically significant effects of an uncertain spatio-temporal structure.

Response: Thanks for the comment. We have now improved the clarity of the core results in the revised version of the manuscript (highlighted in yellow), as well as the interpretation of the synergistic and redundant effects in the Discussion. The main changes are the following:

Line 186-206: *“The dynamics of spatio-temporal synergy in the ERP and BB signals showed complex and heterogenous patterns between early time points of the auditory electrodes and later time points in the frontal electrodes (Figure 4). For example, while the ERP signals encoded both diagonal (Figure 4A; grey clusters ~100-350 ms after tone presentation) and off-diagonal synergistic patterns (Figure 4A; grey clusters ~150-350 ms after tone presentation), the BB signals mainly showed off-diagonal synergy between temporal and frontal electrodes (Figure 4B; grey clusters ~220-350 ms after tone presentation). In Figure 4A, the diagonal stripes suggest the possibility of oscillatory dynamics, where the representation in frontal regions between 50-300 ms is enhanced by knowledge of the activity of temporal regions ~50 ms earlier (the upper diagonal line). Note that 50 ms peak-to-peak timescale corresponds to a frequency of ~10 Hz, i.e. the alpha range. In Figure 4B the off-diagonal block suggests that the frontal representation of the stimulus between 20-120 ms initiates a state change: later temporal activity (200 ms+) enhances the readout of the stimulus class, even though there is no representation of PE in the BB signal of the temporal area at that time.”*

Line 216-224: *“In the Local contrast, although we observed temporal synergy in both ERP and BB signals, the off-diagonal synergy was primarily observed between early and late time points of the BB signals in the temporal cortex (Figure 5B; grey clusters ~150-350 ms after tone presentation). The ERP signals, on the other hand, showed diagonal synergy in both the temporal (Figure 5A; grey clusters ~40-150 ms after tone presentation) and frontal cortex (Figure 5C; grey clusters ~150-350 ms after tone presentation).”*

Line 230-235: *“In the case of the Global contrast, we observed temporal synergy across early and late time points but mostly in the BB signals both within the auditory (Figure 5F; grey clusters ~0-350 ms after tone presentation) and frontal electrodes (Figure 5H; grey clusters ~230-330 ms after tone presentation).”*

Lines 435-445: *“The off-diagonal synergy between early and late time points could be a signature of a neural state shift. It is interesting to note that the synergy remains strong over periods after the PE response is no longer represented (i.e. no MI at those time points). However, the initial representation of the PE may have shifted the local network dynamics into a different state. Then knowing this ongoing state improves the readout of the encoded information at the earlier time point. Thus, the off-diagonal synergy might be an echo of the initial PE representation that is not directly observable in later time points.*

Lines 446-457: *Synergy can also arise from a common source of neural noise that is non-stimulus specific. For example, the spatio-temporal synergy between regions could reflect a global change in attention or arousal. In this situation, the readout of one area provides information about the global neural state even when it doesn't convey information about the PE directly, and this can be used to improve the resolution with which the PE can be decoded from the other area. Although this might be a possibility, the tight timing of the synergy bands observed in both experiments (i.e. diagonal and off-diagonal synergistic patterns) speak more of a transient dynamics rather than global ongoing fluctuations underlying the spatio-temporal synergy.”*

The data from the computational model of brain networks is also only casually (qualitatively) compared to the empirical electrophysiological data, and related statements such as “[the model] accurately replicates critical

neurobiological and neuroanatomical features of the mammalian cortex", "we immediately observed..." etc. are clearly exaggerated.

Response: We are grateful to the reviewer for raising these important points. We should clarify that the statement the Reviewer refers to ("*the model accurately replicates critical neurobiological...*" etc.) is based on a number of previously published modelling results (Garagnani, Lucchese, Tomasello, Wennekers, & Pulvermüller, 2016; Garagnani & Pulvermüller, 2011, 2016; Garagnani, Wennekers, & Pulvermüller, 2008, 2009; Henningsen-Schomers, Garagnani, & Pulvermüller, 2023; Pulvermüller & Garagnani, 2014; Schomers, Garagnani, & Pulvermüller, 2017; Tomasello, Garagnani, Wennekers, & Pulvermüller, 2017, 2018; Tomasello, Wennekers, Garagnani, & Pulvermüller, 2019) obtained using the same architecture to simulate and explain neural mechanisms underlying language acquisition, attention, memory, and automatic auditory change detection (including the MMN response to familiar and unfamiliar sounds (Garagnani & Pulvermüller, 2011)). In brief, the model realises the following physiological and anatomical features:

- Neurophysiological dynamics of pyramidal cells including temporal summation of inputs, action potential propagation, and firing rate adaptation - these reflect well-known properties of neuronal cells of the mammalian cortex (Matthews, 2001);
- Local activity regulation by means of inhibitory interneurons (implemented as inhibitory cells), is known to be pervasive in the cortex (Braitenberg, 1978; Yuille & Geiger, 2003);
- within- and between-area connectivity characterised by sparse, patchy, and topographic links, along with a neighbourhood bias towards close-by links, as observed in the cortex (Amir, Harel, & Malach, 1993; Braitenberg & Schüz, 1998; Gilbert & Wiesel, 1983, 1989; Kaas, 1997);
- Between-area model links that reflect the presence of corresponding neuroanatomical links between homolog cortical areas in the marmoset brain (see section "*Connectivity of the simulated brain areas*" in the Methods for references); and
- Constant uniform uncorrelated white noise in all model neurons, simulating spontaneous baseline neuronal firing, a very well-documented neurophysiological feature of the brain (Rolls & Deco, 2010).

Our approach here, which builds upon a number of previous neurocomputational modelling works (e.g. Garagnani et al, 2008; Garagnani and Pulvermüller, 2011; Garagnani & Pulvermüller 2013; Pulvermüller & Garagnani, 2014; Garagnani & Pulvermüller, 2016; Tomasello et al., 2017; Henningsen-Schomers et al. 2023; Shtyrov *et al.*, 2023), is to take into account as many biological constraints considered essential for implementing neurocomputational models of cognitive and brain function (see Pulvermüller et al. 2021) as practically viable given the available computational resources. While models implementing higher levels of biological realism can always be developed, one should also question whether such more refined models would be able to provide a better understanding of the phenomenon of interest. We should also add that we are not aware of any existing neurocomputational model of the relevant marmoset brain areas that can accurately replicate the observed ECoG responses while implementing all of the above neurobiological constraints; hence, we submit that the present effort is a significant step forward in this direction.

In regard to the point concerning the qualitative comparison between model and ECoG results, we fully agree with the Reviewer: simulation and experimental data should be quantitatively compared; this can be done by means of the respective significance plots reported on the right-hand side of each of the co-information, redundancy, and synergy plots (see the grey-level panels in the figures). Following the Reviewer's suggestion, we now report the Structural Similarity Index (SSIM) values obtained by comparing the co-information charts from the simulations against those obtained from experimental data. The SSIM assesses the structural similarity between two images (see <https://uk.mathworks.com/help/images/ref/ssim.html>), with values ranging from 0 (dissimilar) to 1 (highly similar). Hence, we converted the co-information plots to images, and computed SSIM between them: SSIM scores between simulated and real co-information charts were 0.74 for the temporal cortex (Fig. 7B versus 3B) and 0.83 for the frontal cortex (Fig. 7C vs. Fig. 3D); both of these similarity indexes are significantly above chance level ($p < 0.05$), obtained by re-computing the SSIM index for 1000 randomly shuffled

versions of the images corresponding to the simulated data. We have now added this result to the main text. We thank the Reviewer for this valuable suggestion.

Lines 336-350: “To quantify the similarity of the co-I values between the real and the simulated data, we computed the Structural Similarity Index (SSIM). The SSIM assesses the structural similarity between two images, with values ranging from 0 (dissimilar) to 1 (highly similar). Hence, we converted the co-I plots of the real and simulated data to images and computed SSIM between them. While the SSIM between simulated and experimental co-I was 0.74 (**Figure 7B** versus **Figure 3B**), the frontal cortex comparison showed an SSIM of 0.83 between simulated and experimental co-I (**Figure 7C** versus **Figure 3D**). Both values were significantly above chance level ($p < 0.05$) after comparing them to a distribution of surrogate SSIM values. The surrogate distribution was obtained by computing the SSIM between the experimental co-I image and a shuffled version of the simulated co-I image, and repeating this procedure 1000 times.”

Last, we acknowledge that the statement “We immediately observed...” could be easily misinterpreted, and apologise for this lack of clarity. What we meant to convey here was that the model taken from the (Garagnani & Pulvermüller 2011) study was applied “as is” to the present project, and that, prior to any adjustments/parameter tuning, the network responses already exhibited the presence of synergy; the subsequent process of fine-tuning only improved the fit of the model results with the experimental data from the marmoset, but did not qualitatively change them. We have removed “immediately” and rephrased the statement in question to clarify this. The revised paragraph now reads:

Lines 323-335: “We observed that, before any adjustment of its parameter values, the network already encoded both redundant and synergistic information, specifically, in the signal from its superior-temporal region (including areas A1, AB, PB). We then further constrained the model's dynamics by fine-tuning three of its parameters [...]. This process of parameter tuning [...] did not qualitatively change the network's responses, but simply improved the fit of the responses with the observed data”.

Once again, thank you for identifying this potentially misleading/unclear statement.

References:

- Amir, Y., Harel, M., & Malach, R. (1993). Cortical hierarchy reflected in the organization of intrinsic connections in macaque monkey visual cortex. *J Comp Neurol*, 334(1), 19-46.
- Braitenberg, V. (1978). Cortical architectonics: general and areal. In M. A. B. Brazier & H. Petsche (Eds.), *Architectonics of the cerebral cortex* (pp. 443-465). New York: Raven Press.
- Braitenberg, V., & Schüz, A. (1998). *Cortex: statistics and geometry of neuronal connectivity* (2 ed.). Berlin: Springer.
- Garagnani, M., Lucchese, G., Tomasello, R., Wennekers, T., & Pulvermüller, F. (2016). A Spiking Neurocomputational Model of High-Frequency Oscillatory Brain Responses to Words and Pseudowords. *Front Comput Neurosci*, 10, 145. doi:10.3389/fncom.2016.00145
- Garagnani, M., & Pulvermüller, F. (2011). From sounds to words: a neurocomputational model of adaptation, inhibition and memory processes in auditory change detection. *Neuroimage*, 54(1), 170-181.
- Garagnani, M., & Pulvermüller, F. (2016). Conceptual grounding of language in action and perception: a neurocomputational model of the emergence of category specificity and semantic hubs. *Eur J Neurosci*, 43(6), 721-737. doi:10.1111/ejn.13145
- Garagnani, M., Wennekers, T., & Pulvermüller, F. (2008). A neuroanatomically grounded Hebbian-learning model of attention-language interactions in the human brain. *Eur J Neurosci*, 27(2), 492-513.
- Garagnani, M., Wennekers, T., & Pulvermüller, F. (2009). Recruitment and consolidation of cell assemblies for words by way of Hebbian learning and competition in a multi-layer neural network. *Cognitive Computation*, 1(2), 160-176.
- Henningsen-Schomers, M. R., Garagnani, M., & Pulvermüller, F. (2023). Influence of language on perception and concept formation in a brain-constrained deep neural network model. *Philos Trans R Soc Lond B Biol Sci*, 378(1870), 20210373. doi:10.1098/rstb.2021.0373
- Kaas, J. H. (1997). Topographic maps are fundamental to sensory processing. *Brain Res Bull*, 44(2), 107-112.
- Matthews, G. G. (2001). *Neurobiology: molecules, cells and systems* (2nd Edition ed.): Blackwell Science.
- Pulvermüller, F., & Garagnani, M. (2014). From sensorimotor learning to memory cells in prefrontal and temporal association cortex: a neurocomputational study of disembodiment. *Cortex*, 57, 1-21.
- Pulvermüller, F., Tomasello, R., Henningsen-Schomers, M. R. & Wennekers, T. Biological constraints on neural network models of cognitive function. *Nat. Rev. Neurosci.* 22, 488–502 (2021).

- Rolls, E. T., & Deco, G. (2010). *The Noisy Brain: Stochastic Dynamics as a Principle of Brain Function*. Oxford: Oxford University Press.
- Schomers, M., Garagnani, M., & Pulvermüller, F. (2017). Neurocomputational consequences of evolutionary connectivity changes in perisylvian language cortex. *Journal of Neuroscience*, *37*(11), 3045-3055. doi:10.1523/JNEUROSCI.2693-16.2017
- Tomasello, R., Garagnani, M., Wennekers, T., & Pulvermüller, F. (2017). Brain connections of words, perceptions and actions: A neurobiological model of spatio-temporal semantic activation in the human cortex. *Neuropsychologia*, *98*, 111-129.
- Tomasello, R., Garagnani, M., Wennekers, T., & Pulvermüller, F. (2018). A neurobiologically constrained cortex model of semantic grounding with spiking neurons and brain-like connectivity. *Frontiers in Computational Neuroscience*, *12*, 88.
- Tomasello, R., Wennekers, T., Garagnani, M., & Pulvermüller, F. (2019). Visual cortex recruitment during language processing in blind individuals is explained by Hebbian learning. *Scientific Reports*, *9*(1). doi:10.1038/s41598-019-39864-1
- Yuille, A. L. & Geiger, D. (2003) in *The Handbook of Brain Theory and Neural Networks* (ed. Arbib, M.) 1056–1060 (MIT Press).

Reviewer #2 (Remarks to the Author):

This paper analyzes synergy between different cortical regions and time delays for conveying prediction error signals. The authors report that interactions are primarily synergistic. The analysis is based on event-related potentials (ERP) and broad band (BB) signals. Overall, the results have the potential to provide a useful contribution to the field. However, the observed size of synergy/redundancy is very small, on the order of ~ 0.01 .

Response: We thank the reviewer for their comments, which helped us to improve the manuscript. In the response, we have now performed some simulations, as well as added some discussion to address the reviewer's points.

We appreciate the reviewer's point regarding the effect size. While 0.01 might seem a low number on an absolute scale (i.e. if a reader is more familiar with effect sizes like d'), we don't think that is true in this study. A key advantage of information theory is that it allows a range of quantitative statistical assessments on the same effect size scale of bits. This is the core unit of information – one bit corresponds to a halving of the uncertainty, or alternatively to one yes-no question which splits the space of possibilities into two halves. It's also important to keep in mind that our measure of information in bits, is the average reduction in uncertainty expected on a single trial, from a single sample (MI is calculated per sample). Here the sampling rate is 500 Hz, so 0.01 bits/sample corresponds to around 5 bits/second, which is consistent with some of the highest information-bearing neural signals. Similarly, for comparison, we have used similar MI measures across a range of applications in neuroimaging (Ince et al 2017) including, face vs house contrast in EEG (arguably the strongest categorical ERP contrast) (Rousselet et al, 2014), speech entrainment in MEG (Daube et al. 2019), sampling of visual information with bubbles (Zhan, 2019). In these diverse applications, it is interesting that consistent, robust within-participant effects often fall in the range of 0.01-0.05 bits/sample, similar to what we report here. So we a priori don't agree that these values are very small, although we accept the units may be unfamiliar. We have now added to the Methods the following description:

Line 798-804: *“Note that MI and co-I values are reported in units of bits. A value of 1 bit corresponds to a halving of uncertainty of the trial state when observing the neural response. It is important to keep in mind though that these information values are the average per sample. Here we use a sampling rate of 500Hz, so a value of 0.01 bits/sample corresponds to an approximate information rate of 5 bits/second.”*

References:

- Ince, R.A.A., Giordano, B.L., Kayser, C., Rousselet, G.A., Gross, J. and Schyns, P.G. (2017), A statistical framework for neuroimaging data analysis based on mutual information estimated via a gaussian copula. *Hum. Brain Mapp.*, 38: 1541-1573.
- Rousselet GA, Ince RA, van Rijsbergen NJ, Schyns PG (2014), Eye coding mechanisms in early human face event-related potentials. *J Vis.*, 13:7. doi: 10.1167/14.13.7.
- Daube, C., Ince, R. A., & Gross, J. (2019). Simple acoustic features can explain phoneme-based predictions of cortical responses to speech. *Current Biology*, 29(12), 1924-1937.
- Zhan, J., Ince, R. A., Van Rijsbergen, N., & Schyns, P. G. (2019). Dynamic construction of reduced representations in the brain for perceptual decision behavior. *Current Biology*, 29(2), 319-326.

Importantly, the mutual information values have not been corrected for finite data bias. One approach for correcting for this bias is described here: <https://pubmed.ncbi.nlm.nih.gov/10935917/>. Once the values are corrected for the bias, it is likely that none of the results will be statistically significant.

Response: We estimate all information theoretic quantities with Gaussian Copula Mutual Information. This is a lower bound estimate, which works by fitting a parametric Gaussian copula. The negative entropy of the Gaussian copula is a lower bound on the mutual information between the variables. The Gaussian entropy calculation is based on the determinant covariance matrix of the normalised data, for which an analytic bias

correction exists (it is an analytic correction for the sampling bias estimating the determinant of a covariance matrix from empirical data) (Misra et al., 2005; Goodman, 1963). This is applied for all calculations, so we **do report bias-corrected values.**

References:

Misra, N., Singh, H., & Demchuk, E. (2005). Estimation of the entropy of a multivariate normal distribution. *Journal of multivariate analysis*, 92(2), 324-342.

Goodman, N. R. (1963). The distribution of the determinant of a complex Wishart distributed matrix. *The Annals of mathematical statistics*, 34(1), 178-180.

However, we don't agree that correcting for bias is necessary for reliable inference, or that doing so would affect in any way our non-parametric permutation test results. Bias refers to the estimation error that results when mutual information is estimated from a finite data set. Bias refers to a systematic offset in the the long run expected value of the estimand if the experiment was repeated thousands of times. Non-parametric permutation inference works by shuffling away the relationship of interest in the data, to obtain values from a surrogate null distribution. Because this shuffled data has all other properties inherited from the original data set (dimensionality, number of samples, outliers, etc.) the bias of the information estimates obtained from the permutations should be similar to that of the true calculation. However, it is the variance over permutations that is more important for inference. The measured MI values are then compared to this surrogate permutation distribution. For this procedure, bias correction makes no difference: as it is a constant offset (GCMI bias is a function of dimensionality and number of trials) it shifts both surrogate null values and the true value equivalently, so does not affect the percentile of the surrogate null distribution at which the measured value sits. The following simulation illustrates this:

Examples of permutation inference for a weak (left, mean difference between classes 0.1) and strong (right, mean difference between classes 0.2) effect with (upper panels) and without (lower panels) bias correction. The red line shows the value of the simulated data, blue bars show the results of 1000 permutation shuffles. The percentile of the full value with respect to the permutation distribution is shown in the title. Bias correction shifts both the surrogate null permutation distribution and the measured values, but does not change their relationship. Non-parametric permutation testing and the method of maximum statistics provide a robust approach to inference that accounts for the sample size of the data, as well as any outliers, autocorrelation between samples, etc. and is not affected by bias correction of the mutual information measure.

The code is available here: <https://gist.github.com/robince/de88832a931808f38d137bc1903c81c4>

Please note that the error-bars were not described in figure legends and it is not clear what the shaded regions represent, such as whether these are standard errors of the mean, standard deviation, or 95% confidence intervals.

Response: Thank you for the opportunity to clarify this point. The shaded regions in the MI plots represent the standard error of the mean (S.E.M). We have now clarified this point in all figure legends.

The authors should acknowledge and link better to a larger body of work that considered co-information in computational systems. In particular, <https://www.jneurosci.org/content/23/37/11539> provided detailed discussion of co-information and how it should be interpreted. Regarding predictive information, the work by Israel Nelken should be cited and discussed, e.g. <https://pubmed.ncbi.nlm.nih.gov/22248537/> and <https://www.nature.com/articles/nn1032>

Response: Thanks for the comment. We have incorporated the following changes in the revised version of the manuscript:

Lines 414-425: *“There are a wide variety of ways of using information theoretic, and other measures to study representational interactions in neural coding (Chicharro, 2014). Schneidman et al. (2003) discuss three types of response independence in the context of spiking neuron population coding: activity independence, conditional independence and information independence. Here we focus only on information independence, as we are interested in relating the information representation between areas. Deviations from information independence are best measured with co-information. To date, co-information has been less frequently applied to aggregate signals as we do here (Ince, 2017; Zhan et al., 2019; Ince et al., 2016).”*

References:

Chicharro, D. A Causal Perspective on the Analysis of Signal and Noise Correlations and Their Role in Population Coding. *Neural Comput* 2014; 26 (6): 999–1054.

Schneidman, E., Bialek, W., & Berry, M. J. (2003). Synergy, redundancy, and independence in population codes. *Journal of Neuroscience*, 23(37), 11539-11553.

Ince, R.A.A., Giordano, B.L., Kayser, C., Rousset, G.A., Gross, J. and Schyns, P.G. (2017), A statistical framework for neuroimaging data analysis based on mutual information estimated via a gaussian copula. *Hum. Brain Mapp.*, 38: 1541-1573.

Zhan, J., Ince, R. A., Van Rijsbergen, N., & Schyns, P. G. (2019). Dynamic construction of reduced representations in the brain for perceptual decision behavior. *Current Biology*, 29(2), 319-326.

Ince, R. A., Jaworska, K., Gross, J., Panzeri, S., Van Rijsbergen, N. J., Rousset, G. A., & Schyns, P. G. (2016). The deceptively simple N170 reflects network information processing mechanisms involving visual feature coding and transfer across hemispheres. *Cerebral Cortex*, 26(11), 4123-4135.

Reviewer #3 (Remarks to the Author):

In this exciting paper, Gelens et. al. use two auditory protocols in non-human primates with ECOG to study how prediction error is processed across brain regions. To that end, they computed co-information (a method that distinguishes redundant from synergic information processing). This paper is fairly written; the authors present an impressive number of results that are, at certain points are difficult to integrate for the reader. Overall, I am impressed with the work done by the authors. This work will certainly constitute an important addition to the field. However before recommending the publication, there are some aspects that the authors need to clarify to validate their results and the conclusions entirely.

Response: We thank the reviewer for their positive initial evaluation of our work, and for contributing to improving the quality of our manuscript. We now respond to their comments point by point :

1) My main concern is that in all of the presented analyses, there is a bias towards synergic interactions that the processing of the stimulus itself might not drive it, but the interaction with unrelated ongoing activity. This can be directly observed in the interactions between the baseline period (-100 to 0 ms) and the post-stimulus period (0 to 300 ms) in all cases (experimental data - Figures 3, 4, 5, 6 – and model -Figures 7 and 8-) there is a synergic interaction (blue). In my interpretation, this synergic interaction between stimulus-related and non-related activity is a bias that probably affects the rest of the analysis and needs to be corrected. A potential way to do it is to do a baseline correction of the co-information computation and remove the mean value of these periods from the rest of the analysis. This will probably reduce the quantified synergy (and augment the redundancy). The authors must quantify if the results (and the conclusions) change after this control.

Response: We appreciate the Reviewer's comment, but our interpretation of the results is slightly different. We don't see any meaningful significant synergy with the pre-stimulus period (with the exception of one panel, **5F**). In the unthresholded graphs, there is some blue visible there, reflecting the overall residual bias of the computed estimates towards synergy, but we address this through non-parametric two-sided permutation testing including maximum/minimum statistics to control for multiple comparisons over the multiple pairs of time points considered. Therefore, although there is some weak bias in the raw unthresholded effect size plots, this bias doesn't affect the resulting inference (please see response to Reviewer 2). We don't believe baseline corrected computed information values are a statistically valid thing to do in this case, we prefer to leave the raw effect size maps, which should be interpreted alongside the statistical threshold masks (Taylor et al 2023).

In general, we see it as an advantage that synergy can be present even between regions that do not represent the stimulus alone. Indeed, we interpret this as signalling a functionally relevant change of state, which may not be visible with other analysis methods. For example, consider the synergy shown in **Figure 5B**. These bands indicate that by observing later activity (150-300ms; after the stimulus is presented and when there is no further direct representation of the prediction error), the readout of the initial prediction error during the 50-150ms period is improved. This suggests there is a stimulus-induced change of state – while this state itself does not predict the stimulus, knowledge of this later and ongoing state change improves the readout of the evoked representation.

References:

Taylor, P. A., Reynolds, R. C., Calhoun, V., Gonzalez-Castillo, J., Handwerker, D. A., Bandettini, P. A., ... & Chen, G. (2023). Highlight Results, Don't Hide Them: Enhance interpretation, reduce biases and improve reproducibility. *NeuroImage*, 274, 120138.

2) The authors present a computational model to explore potential explanations for the results. The authors argue that comparing a computational model including feedforward and feedback connections and one with only feedforward alone demonstrates that feedback connections are responsible for the emergence of synergic processing. I'm afraid I have to disagree with the authors that the models themselves can demonstrate any

neuronal mechanism. Models are helpful to interpret data and generate predictions that can be subsequently experimentally tested. The results obtained with the model are not evidence of how the mechanism is implemented in the brain.

Response: We are grateful to the Reviewer for raising this important point, which we think may be the result of a lack of clarity on our part, for which we would like to apologise.

We fully agree with the Reviewer: models are helpful in interpreting data and generating predictions, which should be subsequently validated via experimental testing. In order to interpret our ECoG data, we used a fully neurobiologically constrained neural model, in the sense described in Pulvermüller et al., 2021. Given that the simulation results are the outcome of the combined action of the mechanisms implemented in the model, if the only implemented mechanisms reflect well-documented neurophysiological phenomena (as in the present case), then the simulations illustrate how the model correlates of such brain principles are sufficient to produce the observed modelling results. If the model data replicate the experimental data of interest (see below), the inner workings and components of a brain-constrained model can be used to identify *candidate* neural mechanisms and structures constituting potential *explanans* for the experimental results. Such theory-driven predictions, of course, must be validated via experimental testing. The revised Discussion (the last paragraph of the sub-section “*Interpreting synergistic interactions*”) now states this more clearly:

Line 484-490: “*On the basis of this computational result, we conjecture that the cortical homologues of such jumping links (known to exist between corresponding regions of the marmoset brain, see Methods) may play a similarly crucial role in the emergence of the temporo-frontal synergistic interactions observed in the ECoG data. This prediction awaits further validation by means of experimental testing.*”

We also suspect that a missing qualifier (“*in the model*”) at the end of the last statement of the Results section may have misled our esteemed Reader into thinking that we treated our simulation results as providing evidence of corresponding brain processes (or lack thereof). To avoid any misunderstandings, we have completed that statement, which now reads:

Line 400-404: “*... it is not simply the presence of feedback projections, but specifically of [...] links connecting non-adjacent areas of the processing hierarchy that is needed for synergy to emerge in the model*”.

We hope that this clarifies the matter and once again apologise for the confusion our initial phrasing may have caused.

References:

Pulvermüller, F., Tomasello, R., Henningsen-Schomers, M. R. & Wennekers, T (2021). Biological constraints on neural network models of cognitive function. *Nat. Rev. Neurosci.* 22, 488–502.

In addition, the results presented by authors comparing the two types of models (Figures 7 and 8) are suggestive of a reduction of synergy. Still, there's no quantification of this result (overall, the simulations coming out of the two models look quite similar -panels B/D and C/F in figure 7 and panels B/D in figure 8 -).

Response: We feel that this comment may stem from a misunderstanding. **Figures 7** and **Figure 8** should not be directly compared, as they plot different quantities. While panels (B-C), (E-F) in **Figure 7** plot the results of analysing the model’s temporal and frontal responses *separately*, panels B-D in **Figure 8** plot the results of *co-information* analysis *between* the (simulated) temporal and frontal systems. Hence, panels B-C in **Figure 7** should not be contrasted against panels B-D in **Figure 8** (or against each other), but only compared with corresponding experimental data (e.g., **Figure 3**).

In fact, the results plotted in **Figure 7** (left) are intended to show the fit between the model and experimental data, achieved after fine-tuning the network parameters (i.e., the strength of the neuronal adaptation, the local inhibition, and between-area links, as mentioned in the Results section of the revised version highlighted in yellow). In particular, as explained in the figure's caption, the data from the "Fully-Connected" (FC) model architecture (**Figure 7B-C**) closely resemble those obtained from the analysis of the Broadband ECoG signals (**Figure 3B-D**).

Having achieved a good fit between the model and experimental data, we then used the model to make novel predictions. Specifically, we manipulated the network's connectivity structure to investigate the question of how synergistic information might emerge in it. As explained in the revised version, we ran additional simulations using a "Feed-Forward only" (FF) architecture, in which all of the between-area feedback and within-area recurrent links were removed. This was aimed at establishing whether these model links are critical for the emergence of synergistic information within the (simulated) temporal and frontal systems. The FF architecture is shown in **Figure 7D** (and **Figure 8C**, too). As the Reviewer correctly noted, the data in **Figure 7** (B-C vs. E-F) reveal no major differences across the two architectures. A striking difference between the responses of the FC and FF models, however, does emerge if one analyses the synergistic interactions *between* the (simulated) frontal and temporal model regions (**Figure 8**). Specifically, the synergy data in panels B and D of Fig. 8 (see the plots at the bottom-right corners of the two panels) show that, while synergy levels are significant in the FC model (panel B, bottom-right plot), in the FF model they are not (panel D, bottom-right plot). Also, note that this is a quantitative result: panel D shows that there are zero data points with significant synergy, whereas panel B contains significant synergy levels. Additional simulations obtained with a further version of the architecture implementing only next-neighbour between-area links along with recurrent ones (see **Figure S10A**) produced no synergistic temporal-frontal interactions (**Figure S10D**), even after further parameter tuning.

The above results, taken together, indicate that a 'feedback' flow of information from frontal to temporal systems (still enabled in the **Figure S10A** model through the reciprocal PB-PF connection) and information processing via recurrent (within-area) links – also present in the model of **Figure S10A** – on its own is not sufficient to generate significant synergy between the temporal and frontal (model) systems. Instead, it appears that it is the bi-directional higher-order, or "jumping" (Schomers et al., 2017) links reciprocally connecting non-adjacent areas of the FC network (**Figure 7A**) that are required for significant across-systems synergy to emerge. As we now state in the revised Discussion (sub-section "*Interpreting synergistic interactions*"), on the basis of this computational result we *conjecture* that the cortical homologues of such jumping links (known to exist between corresponding regions of the marmoset brain) may play a similarly crucial role in the emergence of the temporal-frontal synergistic interactions observed in the ECoG data. This prediction awaits further experimental validation.

References:

Schomers, M., Garagnani, M., & Pulvermüller, F. (2017). Neurocomputational consequences of evolutionary connectivity changes in perisylvian language cortex. *Journal of Neuroscience*, 37(11), 3045-3055.

REVIEWERS' COMMENTS

Reviewer #3 (Remarks to the Author):

I appreciate the changes introduced by the authors. I have no further questions.

Reviewer #4 (Remarks to the Author):

I have received this manuscript on synergistic vs redundant inter areal communication as a reviewer post its first round of reviews to assess if the authors had taken the initial reviewers concerns adequately into consideration. After reading the revised manuscript, as well as the correspondence to the previous reviewers I have to assess that the reviewers concerns are mostly not taken into consideration, nor properly adressed. The authors disagree largely with very valid criticism on both the conceptual and methodological aspects of the manuscript and largely add descriptions to the manuscript that do not help the caveats mentioned by the reviewers. Overall the lack of isolation and causative inferences make it very hard to interpret the results. The reviewer would like to emphasize that this is true for many studies in this domain but the claims of the paper are a delineation that rests on assumptions not tested or verified. The authors respond to part of that criticism by demonstrating robustness of their results which is laudable but ultimately wrong assumptions can be robust nonetheless. Lastly I would like to point at the very small effects that make this reviewer cautious how solid these results are and if they would actually replicate.